# KL-Regularized Reinforcement Learning for Generative Modelling is Designed to Mode Collapse

**Anthony GX-Chen**[1]  **Jatin Prakash**[1]  **Jeff Guo**[2]  **Rob Fergus**[1]  **Rajesh Ranganath**[1]
[1]New York University    [2]École Polytechnique Fédérale de Lausanne (EPFL)
`anthony.gx.chen@nyu.edu`

## ABSTRACT

Classical intuitions cast minimizing reverse KL as "mode seeking" and forward KL as "mass covering". In KL-regularized reinforcement learning, however, the regularizer determines *both* the target distribution's shape *and* the divergence being implicitly minimized, making its role more nuanced than simply inducing generic mode-seeking or mass-covering behaviour. Specifically, the target distribution is defined jointly by the reward function, the reference model, the type of regularizer, and the regularization strength. We show that under common settings—such as low regularization strength and equal verifiable rewards—both forward and reverse KL regularization tend to specify target distributions whose mass concentrates on a single high-reward region. Thus, the objective itself *by construction* induces diversity collapse, regardless of the policy optimization algorithm used.

Building on this perspective, we introduce a simple and scalable modification that rescales rewards to induce target distributions assigning substantial probability across *all* high-reward regions. This yields a principled objective that maintains high solution quality while achieving broad reward-mode coverage. Empirically, this approach improves post-training diversity and performance for Large Language Models and Chemical Language Models, and is effective with either forward or reverse KL regularization, while using either naively fails.

## 1 INTRODUCTION

Reinforcement Learning (RL) is the predominant method for post-training foundation models (Ouyang et al., 2022), and the primary way to train generative models in settings where the correct solution is not known *a priori*. At its core, this involves solving a regularized RL (contextual bandits) problem, where a policy is trained to maximize some external reward, while preserving "closeness" to a base policy (as to e.g. preserve coherence). Output diversity of the policy is crucial. In Large Language Models (LLMs), it drives engagement for tasks such as creative writing and free-form conversation. More generally, diversity underlies the generation of new knowledge, enabling the discovery of novel mathematical solutions (Romera-Paredes et al., 2024), cognitive science models (Castro et al., 2025), and novel algorithms and software (Surina et al., 2025; Novikov et al., 2025; Aygün et al., 2025). Furthermore, diversity reflects uncertainty over competing hypotheses, a property fundamental to scientific discovery (GX-Chen et al., 2025). Finally, diversity plays an important role *during training* to drive exploration such that the policy can find and converge to better solutions (Cui et al., 2025).

Yet, current empirical evidence suggests RL post-training improves quality at the cost of diversity (Kirk et al., 2023; Cui et al., 2025). As a response, a number of recent works set out to treat this ailment, with a variety of approaches including explicit diversity rewards (Li et al., 2025), changing the KL regularization (Wang et al., 2023), selecting diverse data (Lanchantin et al., 2025), and count-based exploration bonuses (Song et al., 2025).

In this work, we take a step back to diagnose a more fundamental problem: *does the objective being optimized actually have a solution that is diverse?* We find that with current set-ups, the answer is often "no", even with unlimited compute, high quality data, and perfect optimization. We prove that under very commonly used settings (such as weak KL regularization with varied rewards, or *any* KL

regularization if correct answers have the same rewards but vastly different reference policy supports), the globally optimal solution is often *by construction* unimodal. To accomplish this, we analyze KL-regularized RL through tools from variational inference (VI, Jordan et al. 1999; Ranganath et al. 2014) to find and dissect optimal policies for different choices of KL regularization.

Section 2 provides preliminaries about KL divergences and reminds the reader of the mode-seeking / mass-covering behaviour of minimizing reverse / forward KL at suboptimality. Section 3 studies KL-regularized reward maximization as implicitly minimizing a divergence between the current policy and a *target distribution*. Section 4 further analyzes the *shape* of this target solution. We focus particularly on how this distribution puts mass over high-reward regions—i.e. multimodality in terms of reward modes (Definition 3.5). This allows us to understand even if we perfectly minimize *any* divergence between the policy and the target distribution, the resulting policy will still be non-diverse if the *target distribution is defined to be unimodal*. Finally, Section 5 shows how one can directly construct the target distribution to cover all high-reward modes. We specify one such distribution which puts mass over all high-reward regions above a certain threshold, and show this requires only a small change to current algorithms. Each section is empirically supported with didactic simulations. Finally, we apply our method out of the box to LLMs and chemical language models and find that it works for complex, realistic scenarios.

The main contributions can be summarized as follows,

1. We analyze the role of reverse/forward KL regularization in RL as *both* defining the target distribution *and* the implicitly minimized divergence between policy and target.

2. We show the shape of the target distribution is determined by the regularizer, regularization strength, and relative reward and reference probability magnitudes. This has implications on how the target distribution puts mass over high-reward regions.

3. We show with typical hyperparameters, the target distribution is often constructed to put mass over a single high-reward region, making diversity collapse a natural consequence of correctly solving the regularized RL problem (as currently defined), regardless of algorithm.

4. We derive conditions required for broad reward mode coverage, and use this insight to construct a simple and theoretically principled RL algorithm (two-line pseudocode, Alg. 1) that puts uniform mass over *all* high-reward regions, without any external diversity signals.

## 2 THE KULLBACK-LEIBLER (KL) DIVERGENCE

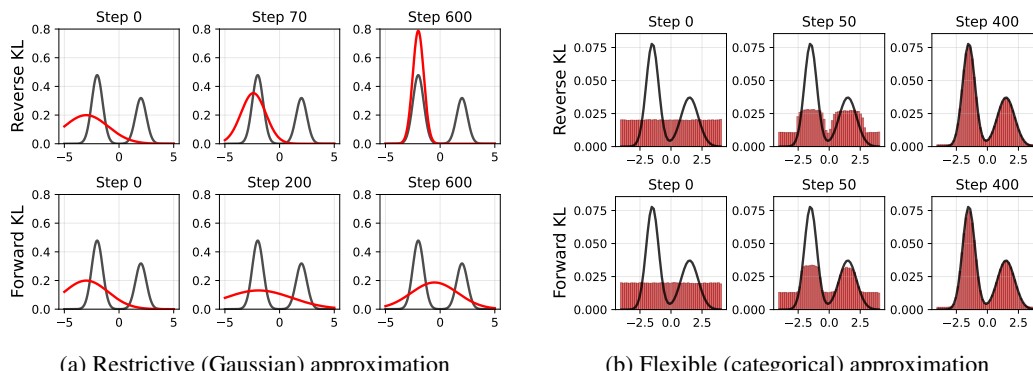

(a) Restrictive (Gaussian) approximation          (b) Flexible (categorical) approximation

Figure 1: Illustration of how the choice of approximate distribution family affects KL optimization. With a restrictive approximate distribution (e.g. 1D Gaussian with two parameters), KL exhibits the typical "mode seeking" and "mass covering" characteristics. This intuition does not necessarily hold for flexible distributions (e.g. independent categoricals, foundational models).

The Kullback–Leibler (KL) divergence (Kullback & Leibler, 1951) measures the discrepancy between two probability distributions. In machine learning, it is commonly used in variational inference (VI), where minimizing the KL divergence enables a tractable variational distribution $q$ to approximate an intractable posterior $p$ (Jordan et al., 1999; Blei et al., 2017). Following Murphy (2012), we refer to $D_{KL}(q||p) = \mathbb{E}_q[\log q(y) - \log p(y)]$ as the *reverse KL divergence*, and $D_{KL}(p||q) = \mathbb{E}_p[\log p(y) - \log q(y)]$ as the *forward KL divergence*. Reverse KL is often described as "mode

seeking", avoiding mass where $p$ is small (Figure 1a, top), while forward KL is often described as "mass covering", putting mass anywhere $p$ has mass (Figure 1a, bottom). These intuitions hold *if* the variational family is not sufficiently expressive and we can at best settle on an optimum with $> 0$ KL (Bishop & Nasrabadi, 2006; Murphy, 2012). With a flexible family, however, optimizing either KLs to the *global optimum* can well-approximate a complex posterior (Figure 1b).

# 3 KL-REGULARIZED REWARD MAXIMIZATION

KL-regularized reward maximization aims to (i) maximize the expected value of a reward function $R : \mathcal{Y} \to \mathbb{R}$, mapping from samples to a scalar outcome (e.g. improve human preference), while (ii) keeping the policy $\pi_\theta$ close to a reference distribution $\pi_{\text{ref}}$ (e.g. maintain grammatical coherence). The objective is $J(\pi_\theta) = \mathbb{E}_{\pi_\theta(y)}[R(y)] - \beta\,D(\pi_\theta, \pi_{\text{ref}})$, where $D(\cdot, \cdot)$ denotes a divergence between the policy and reference distributions. For brevity, we consider the unconditional generation problem where the policy models distribution $\pi_\theta(y)$. The problem is the same in the case of conditional generation (e.g. question answering), where the objective is simply defined over the conditional distribution $\pi_\theta(y|x)$. *We do not deal with the sequential decision making setting*—commonly modelled using Markov Decision Processes (Puterman, 1994)—in this work. Nevertheless, the non-sequential setting is widely used when training generative models with RL.

In this section, we consider the *solution / target distribution* of KL-regularized reward maximization, i.e. the distribution which maximizes the objective. The central question is:

> *If we perfectly solve the regularized RL problem to its global optimum, what does the solution (policy) distribution look like?*

## 3.1 SOLUTION OF THE REVERSE KL REGULARIZED OBJECTIVE

The most common KL-regularized policy gradient objective uses the *reverse KL divergence*,

$$J_\beta(\pi_\theta) = \mathbb{E}_{\pi_\theta(y)}[R(y)] - \beta\,D_{KL}\big(\pi_\theta || \pi_{\text{ref}}\big). \tag{1}$$

A number of previous works have discussed the solution / optimal distribution of this optimization problem (Korbak et al., 2022; Go et al., 2023; Rafailov et al., 2023; Azar et al., 2024; Zhang & Ranganath, 2025), which we note again below (see Appendix B.1 for detailed derivations).

**Remark 3.1.** *The optimal solution to the reverse-KL regularized reward maximization problem,* $\arg\max_{\pi_\theta} J_\beta(\pi_\theta)$*, is given by the target distribution* $\pi^* = G_\beta$,

$$G_\beta(y) = \frac{1}{\zeta}\,\pi_{ref}(y)\exp\Big(\frac{R(y)}{\beta}\Big), \tag{2}$$

*where* $\zeta = \int \pi_{ref}(y)\exp(R(y)/\beta)\,dy$ *is the normalizing constant.*

Remark 3.1 tells us the distribution maximizing Equation 1 is $\pi_\theta = G_\beta$. However, it may not be immediately obvious *how* optimizing Equation 1, $\nabla_\theta J_\beta(\pi_\theta)$, moves $\pi_\theta$ toward $G_\beta$. We analyze this below (details in Appendix B.2, also see e.g. Zhang & Ranganath (2025)).

**Remark 3.2.** *The gradient of Equation 1 is a gradient of the reverse KL divergence between the current policy* $\pi_\theta$ *and the target distribution* $G_\beta$,

$$\nabla_\theta\,D_{KL}\big(\pi_\theta || G_\beta\big) \propto -\nabla_\theta\,J_\beta(\pi_\theta). \tag{3}$$

> *Main Takeaway*
>
> Maximizing the reverse-KL regularized RL objective $J_\beta$ (Equation 1) is equivalent to doing distribution matching by minimizing a reverse KL toward the target distribution $G_\beta$ (Equation 2).

## 3.2 SOLUTION OF THE FORWARD KL REGULARIZED OBJECTIVE

Alternatively, the reward can be maximized with a forward KL penalty,

$$J_{\text{fwd}}(\pi_\theta) = \mathbb{E}_{\pi_\theta(y)}[R(y)] - \beta\,D_{KL}\big(\pi_{\text{ref}} || \pi_\theta\big). \tag{4}$$

A number of recent works have used forward KL regularization. Some are motivated explicitly by the "mass covering" intuition of the forward KL (Wang et al., 2023), while others—such as GRPO (Shao et al., 2024; Guo et al., 2025a)—may have incidentally estimated the forward KL, despite meaning to use the reverse KL (Tang & Munos, 2025).

**Remark 3.3.** *Assume optimization with $\beta > 0$, with finite rewards $R_{max} < \infty$, and there exist solution(s) where $R(y) = R_{max}$, $\pi_{ref}(y) > 0$. The optimal solution to the forward-KL regularized reward maximization problem, $\arg\max_{\pi_\theta} J_{fwd}$, is given by the distribution:*

$$G_{fwd}(y) = \frac{\beta\,\pi_{ref}(y)}{\Lambda - R(y)}, \quad \Lambda > \max_y R(y), \tag{5}$$

*where a unique $\Lambda$ exists for each $\beta$ such that $G_{fwd}$ is a valid probability distribution.*

Notably, Equation 5 is a *completely different* distribution family from the reverse KL case (Equation 2), and does not have a simple closed form unnormalized solution. It is also worth noting that *if* higher-rewarding regions exist outside of $\pi_{ref}$'s support, $G_{fwd}$ *can* place nonzero mass on regions where $\pi_{ref}(y) = 0$ and $R(y) = R_{max}$, with no preference among $y$'s within this region in terms of density. See Appendix B.3 for more details.

**Remark 3.4.** *Assume we are optimizing within the support of $\pi_{ref}$, the gradient of Equation 4 is **not** a forward KL gradient,*

$$\nabla_\theta\, D_{KL}\big(h \,\|\, \pi_\theta\big) \not\propto -\nabla_\theta\, J_{fwd}(\pi_\theta), \tag{6}$$

*for **any** target distribution $h$ that is defined independently of $\pi_\theta$, and arbitrary reward functions $R$.*

*Proof.* Appendix B.4. □

Therefore, while Equation 4 can still be a good objective to optimize, it does not necessarily inherit exactly the same properties and intuitions as a "forward KL gradient".

What, then, is the gradient of the forward KL $D_{KL}(G_\beta\|\pi_\theta)$? It in fact reduces to doing maximum likelihood (e.g. supervised fine-tuning) on trajectories sampled from the target $G_\beta$ (Remark B.2), which is intractable for generic targets. However, this provides one perspective on algorithms such as STaR (Zelikman et al., 2022) and RAFT (Dong et al., 2023; Xiong et al., 2025) that filter high-reward trajectories for maximum likelihood. One can interpret filtering as rejection sampling to approximate a target distribution (which put high mass over high-reward regions), when reward is bounded and we know $G_\beta$ up to normalization. More generally, other methods that approximately sample from $G_\beta$ to minimize divergence include Naesseth et al. (2020); Khalifa et al. (2020).

> *Main Takeaway*
>
> Maximizing the forward-KL regularized objective $J_{\text{fwd}}$ (Equation 4) does not yield a forward-KL gradient, so its behaviour cannot be naively equated to forward-KL optimization.

To summarize: the regularized RL objective implicitly minimizes a divergence between the current policy $\pi_\theta$ and a target distribution $G$. Different choices of regularizer lead to different target distributions $G$. Importantly, the regularizing divergence $D(\pi_\theta, \pi_{\text{ref}})$ need *not* be the same type of divergence as the one implicitly minimized, $D(\pi_\theta, G)$, as is the case for forward-KL regularization.

### 3.3 Both KL Regularization Can Have Multimodal Solution Distributions

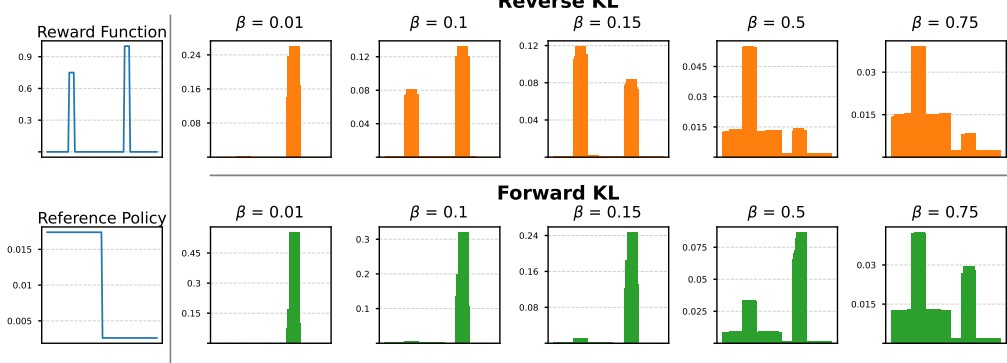

Figure 2: Final policy distribution (100-dim categorical) from optimizing a reverse/forward KL regularized reward maximization objective, given the same reward function, reference policy, across a range of regularization strengths ($\beta$). Both KLs can lead to multimodal solution distributions.

We briefly note that the target distributions for both the reverse (Equation 2) and forward (Equation 5) KL regularization *can* be multimodal. To ground the discussion, we first define a common-sense notion of "multimodal" in terms of reward modes, which we use for the rest of the paper.

**Definition 3.5.** A target distribution $G$ for regularized reward maximization is "**reward multimodal**" given a high-reward threshold $\tau$ if: for any two samples $y$, $y'$, $R(y) < \tau \le R(y')$ implies $G(y) < G(y')$; and for any above-threshold samples $R(y) \ge \tau$ and $R(y') \ge \tau$, $G(y) \approx G(y')$.

Informally, this means all high-reward samples have a high probability, and sampling from $G$ samples approximately equally from *all* high-reward regions.

We show in a didactic example in Figure 2, where given the same reward function containing two high-reward modes, and a reference policy with support over the first half of the action space, optimizing the reverse and forward KL objectives lead to a wide variety of solutions that depend on the regularization coefficient $\beta$. Both KLs have settings of $\beta$ that induce *reward multimodal* target distributions. We analyze the properties of the target distribution in the subsequent section, and return to the Figure 2 example in detail in Section 4.3.

# 4 ANALYSIS OF KL REGULARIZED OPTIMAL DISTRIBUTION

We have seen in Section 3.3 that both KL-regularized RL objectives can have *reward multimodal* solutions, and in Section 2 that optimizing either KL divergence to global optimum will give us policies that well-approximate the (multimodal) solution. However, the shape of the target distribution depends on the reward, reference distribution, and regularization strength. This raises the central question:

> *Is the globally optimal solution we commonly define in KL-regularized RL actually reward multimodal (Definition 3.5)?*

The central tool we use in this section is a *probability ratio* between two samples under a distribution. Intuitively, we want (i) high-reward samples to be much more probable than low-reward samples, and (ii) similarly high-reward samples to have similar high probabilities. Unless otherwise stated, we focus our analysis on the solution of the reverse-KL regularized objective (Equation 2), both for its clean form and because it is the most common way KL-regularized RL is formulated.

**Proposition 4.1.** *The (log) probability ratio between any two samples, $y_1$, $y_2$, under the optimal solution distribution for reverse-KL regularized RL, $G_\beta$, can be written in closed form,*

$$\log \frac{G_\beta(y_1)}{G_\beta(y_2)} = \log \frac{\pi_{ref}(y_1)}{\pi_{ref}(y_2)} + \frac{1}{\beta}\Big(R(y_1) - R(y_2)\Big). \tag{7}$$

*Proof.* Because normalization constant $\zeta$ cancel out in ratios. See Appendix B.6. $\square$

Proposition 4.1 gives us a generic and closed-form way of analyzing how likely one sample is relative to another in the *optimal solution*, using *only* $\pi_{\text{ref}}$ and the reward function $R$, for *any* reverse-KL regularized reward maximization objective. This gives us a number of consequential insights.

## 4.1 WITH EQUAL SUPPORTS, SMALL REWARD CHANGE DRIVES LARGE PROBABILITY CHANGE

**Remark 4.2.** *For any two samples $y_1$ and $y_2$, if $\pi_{ref}(y_1) = \pi_{ref}(y_2)$, their probability ratio is:*

$$\frac{G_\beta(y_1)}{G_\beta(y_2)} = \exp\Big(\frac{R(y_1) - R(y_2)}{\beta}\Big). \tag{8}$$

We first consider the continuous reward function setting where samples have small differences in rewards. This is common for settings such as alignment (reward model) and drug discovery (e.g. binding affinity). If two samples have the *same* probability under the reference distribution $\pi_{\text{ref}}$ ("equal support"), the difference in their final log probabilities is simply the difference in their rewards, scaled by $1/\beta$. Smaller $\beta$ exaggerates the difference

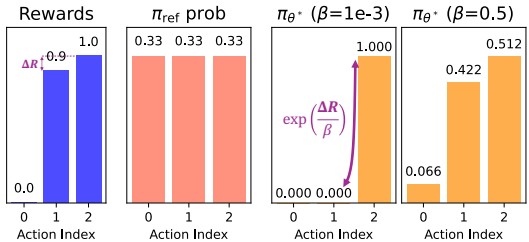

Figure 3: With equal $\pi_{\text{ref}}$, linear difference in rewards ($\Delta R$) lead to exponential difference in probabilities

between relative probabilities. Note a *linear* difference in rewards result in an *exponential* difference in probabilities: for a 0.1 difference in rewards, and a commonly used $\beta = $ 1e-3, the higher reward sample is pushed to be $2.6 \times 10^{43}$ times more likely in the solution distribution. This issue is identically present for entropy-only regularization (see Fig. 7 for effect of $\beta$ on relative probabilities). This suggests for commonly used hyperparameters, the solution is highly concentrated around the max reward mode(s).

We see in Figure 3 a didactic experiment that verifies this theory. At low regularization strength ($\beta$), the optimized policy $\pi_{\theta*}$ mode collapses onto the highest reward action. At high $\beta$, policy achieves better (still not perfect) coverage over high-reward answers, at the cost of having more mass on low reward actions (more details and results in Appendix C.1).

## 4.2 WITH EQUAL REWARDS, SOLUTION NEVER PREFERS LOWER-SUPPORT SAMPLES

We now consider the case where the correct solutions all have *equal* reward. This is a standard set-up for RL with verifiable reward (e.g. math), where a correct answer is given a reward of 1, and incorrect answers given 0.

**Remark 4.3.** *For any two samples with the same reward, $R(y_1) = R(y_2)$, their probability ratio is:*

$$\frac{G_\beta(y_1)}{G_\beta(y_2)} = \frac{\pi_{ref}(y_1)}{\pi_{ref}(y_2)}. \tag{9}$$

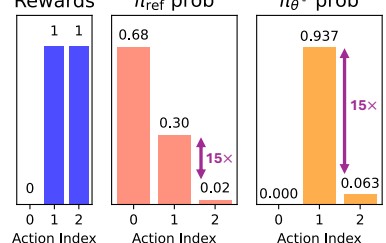

Figure 4: With equal rewards, RL does not change answers' relative probs.

In words, the correct answers' probability ratio in the optimal solution is simply their probability ratio in the reference distribution $\pi_{\text{ref}}$. This ratio is *independent* of the regularization strength $\beta$. In other words, by construction, KL-regularized RL with equal rewards *never promotes a low-support answer*.[1] Figure 4 demonstrates this point empirically: the final policy *never* favours the equally correct low-support mode (additional results in Appendix C.1). This is not an issue with exploration; we will see in the subsequent section that with a small change in reward one can optimize for a distribution that equally weights or even prefers the lower-support solution.

> *Main Takeaway*
>
> RL with *any* KL-regularization does not increase the relative probability of lower-support samples to high-support ones, as long as their rewards are the same. Lowering the KL regularization strength $\beta$ has *no effect* on up-weighting low-support samples in the optimal solution.

We additionally corroborate in Appendix C.4 that in practice for LLMs, the shape of the reference distribution and reward function does result in highly skewed target distribution $G_\beta$, per Proposition 4.1, Remark 4.2, and Remark 4.3.

## 4.3 FOR UNEQUAL REWARDS *and* SUPPORTS, REGULARIZATION STRENGTH DETERMINES MODE COVERAGE

When two trajectories have different rewards and different probabilities under the reference policy, a unique setting of $\beta$ will induce the two to have the same probability in the target distribution.

**Remark 4.4.** *Two samples have the same probability in the target distribution if,*

$$R(y_2) - R(y_1) = \beta\big(\log \pi_{ref}(y_1) - \log \pi_{ref}(y_2)\big). \tag{10}$$

This condition allows us to predict, given only the reward and reference policy, when two samples will have the same probabilities in the solution to the RL problem. As an example, we know in Figure 2 that the two high-reward modes have rewards 0.75 and 1.0, and reference policy probabilities of $\log \pi_{\text{ref}}(y_1) \approx -4.05$ and $\log \pi_{\text{ref}}(y_2) \approx -5.95$, respectively. This allows us to predict the setting of $\beta$ which will "flip" the target distribution's preference from the high-support mode to the low-support mode to be $(1 - 0.75)/(-4.05 + 5.95) \approx 0.132$. Indeed, we see in Figure 2 for the reverse KL case, the preference between the two modes switch as we move from $\beta = 0.15$ to $\beta = 0.10$. This is the true role of the regularization coefficient $\beta$: it is a knob that decides between picking higher rewarding, low-support solutions, vs. lower rewarding, high-support solutions.

---

[1]This observation is true for both reverse and forward-KL regularized RL.

## 5 Directly Optimizing a Reward Multimodal Target

Having identified the various failure cases of the KL-regularized RL objective (Section 4), and the role of regularization in balancing reward differences (Section 4.3), we now turn to the question:

*Can we construct an objective that, when optimized, naturally gives rise to a reward multimodal target distribution?*

Indeed, Remark 4.4 already provides the equality condition required to achieve this. We derive a simple procedure which will ensure we are optimizing for a solution that puts *equal* probabilities on all high-quality samples (per Definition 3.5), using the augmented reward function,

$$\bar{R}(y) = \begin{cases} R(y) & \text{if } R(y) < \tau, \\ R(z) + \beta\big(\log \pi_{\text{ref}}(z) - \log \pi_{\text{ref}}(y)\big) & \text{if } R(y) \geq \tau, \end{cases} \tag{11}$$

where $\tau \leq \max_y R(y)$ is some threshold for "goodness", and $z$ is a fixed "anchor" sample chosen from the set of high-quality samples. We can pick it to be $z = \arg\max_y \pi_{\text{ref}}(y)$ where $R(y) \geq \tau$. Because we are choosing the "anchor" to be from a high-reward mode, we colloquially refer to this as "*mode anchoring*", and the method as *Mode Anchored Reward Augmentation* (**MARA**). See Algorithm 1 for pseudocode with minimal changes (an alternative that augments reward and $\pi_{\text{ref}}$ is outlined in Algorithm 2, which is equivalent to Alg.1 when using reverse KL regularization).

---

**Algorithm 1** Mode Anchored Reward Augmentation (MARA), within a sampled batch.
Changes from a standard RL algorithm are in blue.

---

1: Given: initial policy $\pi_\theta$, reference distribution $\pi_{\text{ref}}$, reward function $R$, regularization coefficient $\beta$, threshold of good answers $\tau \in \mathbb{R}$, $\tau \leq \max_y R(y)$, and trajectory batch $\{y_i\}_{i=1}^N \sim \pi_\theta$.
2: Pick anchor trajectory: $z = \arg\max_{y_i} \pi_{\text{ref}}(y_i)$, s.t. $R(y_i) \geq \tau$
3: **for** each $y_i$ in batch **do**
4:    **if** $R(y_i) \geq \tau$ **then**
5:       Augment: $\bar{r}_i = R(z) + \beta\big(\log \pi_{\text{ref}}(z) - \log \pi_{\text{ref}}(y_i)\big)$
6:    **else**
7:       Keep same: $\bar{r}_i = R(y_i)$
8:    **end if**
9: **end for**
10: Optimize policy parameters $\theta$ using augmented rewards $\{\bar{r}_i\}_{i=1}^N$.

---

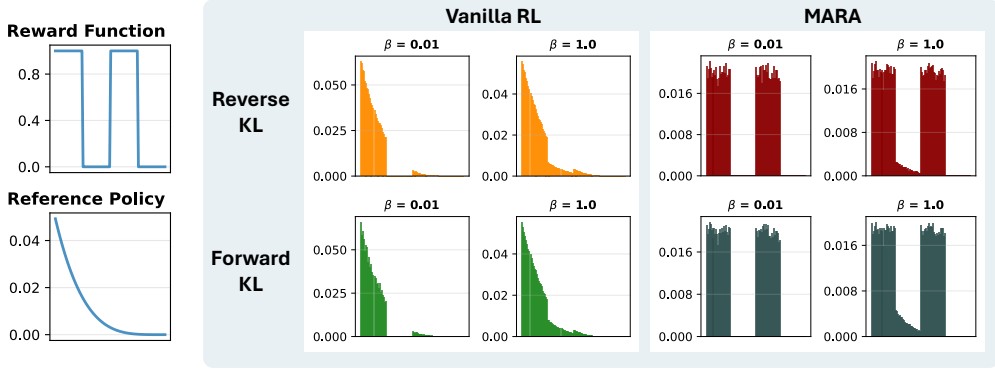

Figure 5: MARA stays close to the reference policy in low-reward areas, and puts high, uniform mass over all high-reward areas.

Intuitively, the augmented reward function constructs a new *target distribution* with *uniform* high density over regions of high reward, and stays close to the reference $\pi_{\text{ref}}$ in regions of low reward (see Remark B.3 for an analysis of the shape of the MARA target distribution). We see in the Figure 5 that vanilla KL-regularized RL result in a policy that heavily favours the left (on-support) mode, regardless of the choice of $\beta$ or KL. On the other hand, using MARA results in solutions that put *equal* high mass over *all* high quality samples, for both KLs. Note that in cases where the reward function range is known, one can directly set threshold $\tau$ as a constant. If not, one can set $\tau$ on a per-batch basis by e.g. taking an upper percentile of sampled rewards (as we do below for non-verifiable LLM tasks).

# 6 EMPIRICAL VALIDATIONS

We evaluate MARA as a drop-in method in a variety of post-training tasks. While our theory has mainly been about the final optimal solution RL achieves, we empirically investigate whether training, even if stopped early, can still benefit from a more diverse global optimum. To do this, we evaluate MARA in (i) verifiable LLM task with multiple answers, (ii) non-verifiable task with reward models, and (iii) chemical language model task for drug discovery, where mode collapse is detrimental.

## 6.1 VERIFIABLE 1-2 TASK FOR LLM

We train an LM (`Qwen2.5-3B`) to generate uniform random integers that are either 1 or 2. It gets a reward of 1.0 for correct (producing "1" or "2" in XML), and 0.0 otherwise (details in Appendix C.2). Most runs are able to optimize the reward well and achieve a reward of approximately 1 (Figure 6a, right). Figure 6a (left) shows the number of correctly formatted 1's the LM generates over the course of training. We see that for naive KL regularization (grey), across a range of $\beta$'s and seeds, all but one run collapse into generating only a single answer as a result of RL, and most collapse into generating 1's, which has higher likelihood under the base policy. MARA (blue), on the other hand, is able to preserve the diversity in the correct answers, with many runs learning to generate 1's and 2's with near uniform probability, while still learning to generate with the correct format (Figure 6a, middle). Further, the Pareto front of model checkpoints at different points in training shows that for both reverse and forward KL regularization, MARA is able to match vanilla training in terms of correctness, while exceeding vanilla training in terms of generation diversity.

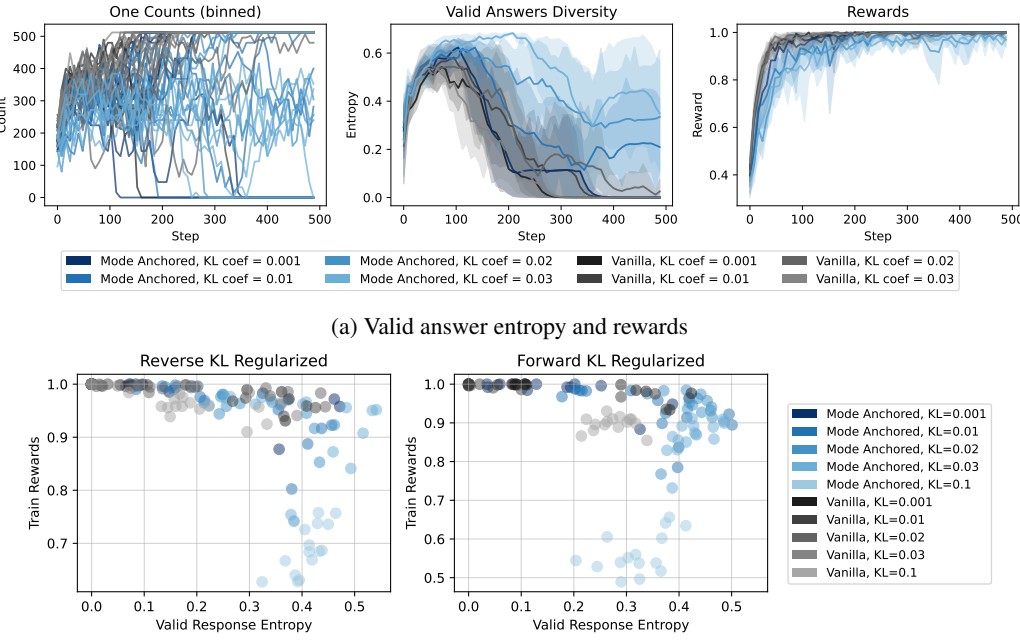

(a) Valid answer entropy and rewards

(b) Pareto front of reward (quality) and entropy (diversity)

Figure 6: Performance on verifiable task with multiple solutions, against both reverse & forward KL. MARA (blue) is compared against baseline GRPO (grey) at different $\beta$'s (`KL coef`).

## 6.2 CREATIVE QUESTION ANSWERING FOR CHAT LLM

We test MARA in a non-verifiable alignment task. We train `Qwen3-1.7B` on a subset of WildChat text (Zhao et al., 2024), using a parametric reward model (`Skywork-Reward-V2-Qwen3-4B`). We evaluate the model on a curated test set (Zhang et al., 2025) and report both the training reward (`In dist Reward`), and test set reward from a different reward model (`Out dist Reward`, using `Skywork-Reward-Gemma-2-27B-v0.2`). We also report diversity metrics in terms of n-grams (`Ngrams`), semantic embeddings (`Semantic Div`), and "distinct functional classes" (`Mean Distinct`). See Appendix C.3 for more details. Here, MARA is used as a drop-in replacement in an RLOO style algorithm (Ranganath, 2017; Kool et al., 2020; Ahmadian et al., 2024).

As the reward function range is not known beforehand, we set $\tau$ on a per-batch basis as the 90th percentile reward of each batch (we ablate this in Appendix Table 5 to find that lowering the percentile decreases performance slightly, but nevertheless remains competitive). We compare against regular RL training, and a number of diversity-promoting baselines including entropy regularization (`Entropy`), rewarding the unlikely (`Unlikely`, He et al. 2025), best-of-N training (`BoN Training`, Tang et al. 2025), and weight ensembling (`Ensemble`, Dang et al. 2025). We see MARA out-performs all baselines in terms of out-of-distribution rewards, and all-but-one diversity metrics (Table 1).

| Model | In-dist. Reward ($\uparrow$) | Out-dist. Reward ($\uparrow$) | Ngrams EAD ($\uparrow$) | Semantic Div ($\uparrow$) | Mean Distinct ($\uparrow$) |
|---|---|---|---|---|---|
| Base Model | 10.94 | 1.166 $\pm0.076$ | 0.413 $\pm0.015$ | 0.220 $\pm0.009$ | 4.01 $\pm0.254$ |
| GRPO | 14.80 | 1.317 $\pm0.102$ | 0.497 $\pm0.014$ | 0.193 $\pm0.009$ | 3.96 $\pm0.249$ |
| RLOO | 15.56 | 1.280 $\pm0.100$ | 0.514 $\pm0.014$ | 0.192 $\pm0.008$ | 3.88 $\pm0.243$ |
| Entropy | 1.44 | 0.786 $\pm0.073$ | 0.267 $\pm0.009$ | **0.228** $\pm\mathbf{0.009}$ | 3.45 $\pm0.193$ |
| Unlikely | 10.04 | 1.381 $\pm0.114$ | 0.532 $\pm0.015$ | 0.191 $\pm0.008$ | 4.24 $\pm0.239$ |
| BoN Training | **16.88** | 0.596 $\pm0.055$ | 0.541 $\pm0.010$ | 0.162 $\pm0.008$ | 2.29 $\pm0.173$ |
| Ensemble | – | 1.143 $\pm0.086$ | 0.438 $\pm0.014$ | 0.211 $\pm0.010$ | 4.19 $\pm0.269$ |
| MARA (rev) | 15.42 | 1.451 $\pm0.103$ | 0.543 $\pm0.014$ | 0.186 $\pm0.008$ | 4.14 $\pm0.233$ |
| MARA (fwd) | 15.33 | **1.604** $\pm\mathbf{0.113}$ | **0.568** $\pm\mathbf{0.012}$ | 0.193 $\pm0.009$ | **4.62** $\pm\mathbf{0.258}$ |

Table 1: Performance on non-verifiable creative task. Mean $\pm$ bootstrap SEM.

### 6.3 DRUG DISCOVERY WITH CHEMICAL LANGUAGE MODELS

Finally, we apply MARA to a distinctively different domain where diversity and quality is crucial: drug discovery. Chemical language models (CLMs) have seen success in discovering molecules in clinical trials. We adapt two realistic reward functions from Guo et al. (2025b): `SYNTH` and `ALL-AMIDE` that jointly reward binding potency and synthesizability. The core CLM optimization problem is also a regularized RL problem: maximize expected reward, while staying close to a pretrained "prior" model to ensure chemical validity. Unlike the traditional RL setting, CLMs are evaluated based on their ability to generate *unique* molecules (`Yield`) given a *fixed* number of reward function evaluations (which are expensive simulations and/or experiments), making diversity an essential quality for any performant CLMs. The `REINVENT` algorithm (Olivecrona et al., 2017; Guo & Schwaller, 2024b) is a state-of-the-art RL-based method on standard benchmarks (Gao et al., 2022). It optimizes the following objective,

$$\mathcal{L}(\theta) = -\Big[ \log \pi_\theta(y) - \big( \log \pi_{\mathrm{ref}}(y) + \sigma R(y) \big) \Big]^2, \quad y \sim \pi_\theta . \tag{12}$$

which is equivalent to KL-regularized reward maximization. We apply MARA as a *drop-in replacement* to its rewards. Additional evaluation details are in Appendix C.5.

Table 2 shows MARA consistently results in higher average `Yield` (number of *unique* high-reward molecules discovered), and lower `OB100` (efficiency in finding high-reward molecules, measured by reward function calls). The screening level (`Screen`) is the reward threshold above which we accept discovered molecules. Setting MARA's $\tau$ equal to the screening level always results in the highest yield, consistent with MARA's target distribution having uniform density in areas where $R(y) > \tau$. Going further, we also assess "global" diversity (which MARA does not explicitly optimize for) in terms of `IntDiv1` and `#Circles`. Both define more macroscopic differences based on molecular sub-structures. We find MARA is competitive with the baseline here. Overall, we see MARA further boosts REINVENT's optimization efficiency, while maintaining diversity.

## 7 CONCLUSION

In this work, we provide an in-depth understanding of the KL-regularized RL objective, particularly in terms of its diversity. We summarize the main take-aways below.

- Studying the divergence being *implicitly minimized* between the policy and target distribution, $D(\pi_\theta, G)$, is more meaningful for understanding optimization behaviour than studying the divergence of the regularizer, $D(\pi_\theta, \pi_{\mathrm{ref}})$, alone.

| Screen | Algorithm | Yield (↑) | OB100 (↓) | IntDiv1 (↑) | Circles (↑) |
|--------|-----------|-----------|-----------|-------------|-------------|
| 0.80 | REINVENT | $6569 \pm 186$ | $1042 \pm 66$ | $0.766 \pm 0.011$ | $67 \pm 3$ |
| | MARA ($\tau$=0.80) | $\mathbf{6834 \pm 78}$ | $1015 \pm 55$ | $0.761 \pm 0.009$ | $59 \pm 8$ |
| | MARA ($\tau$=0.85) | $6584 \pm 231$ | $1042 \pm 66$ | $0.761 \pm 0.008$ | $72 \pm 6$ |
| 0.85 | REINVENT | $1614 \pm 407$ | $4114 \pm 109$ | $0.701 \pm 0.018$ | $7 \pm 1$ |
| | MARA ($\tau$=0.80) | $1796 \pm 210$ | $\mathbf{3654 \pm 272}$ | $0.716 \pm 0.015$ | $6 \pm 1$ |
| | MARA ($\tau$=0.85) | $\mathbf{2196 \pm 394}$ | $4010 \pm 297$ | $0.703 \pm 0.011$ | $7 \pm 1$ |

(a) SYNTH task

| Screen | Algorithm | Yield (↑) | OB100 (↓) | IntDiv1 (↑) | Circles (↑) |
|--------|-----------|-----------|-----------|-------------|-------------|
| 0.80 | REINVENT | $5433 \pm 184$ | $1427 \pm 63$ | $0.768 \pm 0.012$ | $35 \pm 1$ |
| | MARA ($\tau$=0.80) | $5635 \pm 249$ | $1407 \pm 123$ | $0.766 \pm 0.008$ | $36 \pm 3$ |
| | MARA ($\tau$=0.85) | $5502 \pm 309$ | $1426 \pm 63$ | $0.769 \pm 0.006$ | $34 \pm 3$ |
| 0.85 | REINVENT | $1098 \pm 88$ | $4360 \pm 257$ | $0.721 \pm 0.016$ | $8 \pm 1$ |
| | MARA ($\tau$=0.80) | $\mathbf{1235 \pm 130}$ | $\mathbf{3943 \pm 303}$ | $0.733 \pm 0.009$ | $8 \pm 1$ |
| | MARA ($\tau$=0.85) | $\mathbf{1438 \pm 126}$ | $4230 \pm 401$ | $0.725 \pm 0.008$ | $8 \pm 1$ |

(b) ALL-AMIDE task

Table 2: Results for different tasks and screening levels (`Screen`, higher meaning more strict) for two challenging drug discovery tasks. Error bars ($\pm$) denote standard deviation over 5 independent seeds. Bold indicates if the performance is statistically significantly better than the alternative method for that screening level (one-sided student's t-test, $p < 0.05$).

- The regularizer, $\beta$, and reward function together define the *target distribution* which $\pi_\theta$ optimizes towards. Forward and reverse KL regularizers define different target distributions.

- For common hyperparameters and reward functions used in practice, the target distribution is often *defined to be unimodal*. Thus, perfectly solving the regularized RL objective yields non-diverse optimal policy distributions.

- This diversity loss can be fixed by instead defining multimodal target distributions, such as through dynamically augmenting the reward via MARA.

There are a number of exciting future directions to improve MARA. For one, MARA requires setting $\tau$ (the reward threshold above which the target distribution puts uniform probability mass). The choice of $\tau$ is obvious in some settings (Section 6.1 & 6.3), but is a hyperparameter in settings with unbounded reward functions and unknown a priori thresholds (Section 6.2). We found that setting batch-specific $\tau$ helps, but better approaches may be possible. It should be noted that setting an *overly high $\tau$* is harmless: if no samples meet the $\tau$ threshold, the MARA mechanism does not kick in, and learning reduces to standard RL.

Further, MARA shapes the target distribution to have multimodality, but does not guarantee faster convergence to the target. Having general algorithms to more efficiently reach the target, *or to guarantee distributional properties (e.g. multimodality) at sub-optimality, would be of general importance and complement MARA's target distribution. Further, MARA introduces *one* such target distribution which places uniform mass at regions where $R(y) \geq \tau$. This may not be an optimal choice for all tasks, and considering alternative mode-preserving target distributions can be interesting future work. All in all, we emphasize that regularized RL—as commonly used for generative model training—is inherently a distribution matching problem and should be viewed as such. Rather than relying on intuitions (e.g. about regularizers), we should directly specify distributions with properties we wish to have as the target of policy optimization.

## ACKNOWLEDGEMENTS

This work is supported by ONR MURI #N00014-22-1-2773, ONR #N00014-21-1-2758, and the National Science Foundation under NSF Award 1922658. This work was partly supported by

the NIH/NHLBI Award R01HL148248, NSF Award 1922658 NRT-HDR: FUTURE Foundations, Translation, and Responsibility for Data Science, NSF CAREER Award 2145542, ONR N00014-23-1-2634, NIH R01CA296388, NSF 2404476, Optum, and Apple. Additional support was provided by a Fellowship from the Columbia Center of AI Technology. This work was also supported by IITP with a grant funded by the MSIT of the Republic of Korea in connection with the Global AI Frontier Lab International Collaborative Research. A.GXC. is supported by the Natural Sciences and Engineering Research Council of Canada (NSERC), PGSD3-559278-2021. *Cette recherche a été financée par le Conseil de recherches en sciences naturelles et en génie du Canada (CRSNG), PGSD3-559278-2021.* J.G. is supported by the Natural Sciences and Engineering Research Council of Canada (NSERC), PGSD-521528389. We are particularly thankful for fruitful discussions with Aram-Alexandre Pooladian and Mark Goldstein in the early stages of this project that helped shape the theoretical foundations without which this project would not be possible. We additionally thank Anirudh Buvanesh and Manya Wadhwa for helpful discussion and feedback.

## REPRODUCIBILITY STATEMENT

We use open-source, publicly available libraries for all experimental code. Didactic experiments are constructed in PyTorch (Paszke et al., 2019). Reinforcement learning on LLM training is done using the `nano-aha-moment` (Kazemnejad et al., 2025) and verl (https://github.com/volcengine/verl) github repos. Chemical language model experiments use the official `saturn` github repo (Guo & Schwaller, 2024b). We provide detailed experimental information in Appendix C. Pseudo-code is provided in Algorithm 1 and Algorithm 2.

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

## A    RELATED WORK

**Entropy collapse in RL**    There is a growing line of empirical works observing RL training collapses the diversity in generation output of the resulting post-trained policy (Kirk et al., 2023; Huang et al., 2024; O'Mahony et al., 2024; Cui et al., 2025; Yang & Holtzman, 2025; Yun et al., 2025; Shypula et al., 2025; West & Potts, 2025; Zhao et al., 2025; Dang et al., 2025; Song et al., 2025), such as in formats (Zhao et al., 2025), random generation and creativity (West & Potts, 2025), as well as exploration and reasoning (Cui et al., 2025; Dang et al., 2025; Song et al., 2025). The observations have mostly been empirical.

A few attempts have been made to theoretically understand entropy collapse. Cui et al. (2025) analyzes what per-step policy gradient (approximately) does to the entropy of a tabular softmax policy, and finds that entropy decreases if there is a strong a strong positive correlation between the action probabilities and corresponding advantage values. Dang et al. (2025) analyzes a special case of multi-arm bandits with K equally good arms and a bad arm, and finds that the optimal probabilities correspond to the re-normalized reference probabilities of just the good arms. We note this is a special case of our Remark 4.3.

Generally speaking, entropy preservation via regularization is often referred to as "max entropy RL", with numerous seminal and ongoing works (Ziebart et al., 2008; Haarnoja et al., 2017; Huang et al., 2019). Our analysis contributes to further understanding of this setting by shedding light on the multimodality of the target distribution when doing entropy regularization.

**Training for diversity**    A number of attempts have been made to empirically address diversity collapse. Wang et al. (2023) generalizes the DPO objective (Rafailov et al., 2023) from reverse-KL regularized to a more general class of $f$-divergence regularizers, with the key motivation being that reverse-KL can be mode-seeking, therefore reduce diversity. We argue in this work that the full story is more nuanced and is better analyzed through the *target distribution*. Cui et al. (2025) proposes to directly regularize the update of high-covariance tokens. Cheng et al. (2025) incorporates an entropy term in the advantage to encourage better reasoning. Wang et al. (2025) show that focusing gradient updates on a minority of high-entropy "forking" tokens can improve reasoning. He et al. (2025) proposes a rank-based "unlikeliness" reward, where more likely samples (under current policy) receives a larger multiplicative penalty to the reward. Similarly, Yao et al. (2025) uses token entropy to encourage diversity. Song et al. (2025) proposes a count-based exploration bonus that more highly rewards less frequently seen outcomes (in previous samples), and Hamid et al. (2025) proposes a similar batch-wise reward. Xu et al. (2023) proposes a more inclusive approach to preference alignment. Dang et al. (2025) found that combining weights of earlier and later checkpoints can improve pass@k performance—one specific measure of diversity.

A number of works attempt to directly optimize for diversity. This relies on the existence of additional information that tells us if two samples are different and by how much. In this vein, diverse DPO Lanchantin et al. (2025) and variants (Chung et al., 2025; Ismayilzada et al., 2025) encourage diversity in preference learning by selecting diverse positives/negatives. Similarly related is Li et al. (2025), which use an external model to evaluate diversity (via a semantic classifier) and use the diversity metric to modify the reward. Hamid et al. (2025) proposes to optimize a batch-level objective that is modified by a diversity function. We do not require an external model to evaluate diversity.

In the unregularized setting, Jhaveri et al. (2025) optimizes for an unregularized policy with specific distributional properties. More distantly, GFlowNets also provide diversity-seeking policies that are specifically designed to sample proportionally to reward, albeit they use different algorithms than the KL-regularized policy gradient which is the most commonly used algorithm for LM post-training (Hu et al., 2023; Kwon et al., 2024; Tiapkin et al., 2024).

## B    MATHEMATICAL DERIVATIONS

### B.1    TARGET DISTRIBUTION OF REVERSE-KL REWARD MAXIMIZATION

**Proof of Remark 3.1**    We provide a proof for the maximizer of the generalized reverse-KL and entropy regularized reward maximization objective,

$$J_{\beta,\eta}(\pi_\theta) = \mathbb{E}_{\pi_\theta(y)}[R(y)] - \beta \, D_{KL}(\pi_\theta||\pi_{\text{ref}}) + \eta H(\pi_\theta). \tag{13}$$

The solution $\arg\max_{\pi_\theta} J_{\beta,\eta}$ has the un-normalized form,

$$G_{\beta,\eta}(y) \propto g_{\beta,\eta}(y) = \pi_{\text{ref}}(y)^{\frac{\beta}{\beta+\eta}} \exp\left(\frac{R(y)}{\beta+\eta}\right). \tag{14}$$

*Proof.*

$$J_{\beta,\eta}(\pi_\theta) = \mathbb{E}_{\pi_\theta(y)}\Big[R(y) - \beta\big(\log\pi_\theta(y) - \log\pi_{\text{ref}}(y)\big) - \eta\log\pi_\theta(y)\Big], \tag{15}$$

$$= -(\beta+\eta)\,\mathbb{E}_{\pi_\theta(y)}\left[\log\pi_\theta(y) - \big(\frac{R(y)}{\beta+\eta} + \frac{\beta}{\beta+\eta}\log\pi_{\text{ref}}(y)\big)\right], \tag{16}$$

$$= -(\beta+\eta)\,\mathbb{E}_{\pi_\theta(y)}\left[\log\pi_\theta(y) - \log\pi_{\text{ref}}(y)^{\frac{\beta}{\beta+\eta}}\exp\big(\frac{R(y)}{\beta+\eta}\big)\right], \tag{17}$$

$$= -(\beta+\eta)\,\mathbb{E}_{\pi_\theta(y)}\left[\log\pi_\theta(y) - \log G_{\beta,\eta}(y)\right] + (\beta+\eta)\log\zeta_{\beta,\eta}, \tag{18}$$

$$= -(\beta+\eta)D_{KL}\big(\pi_\theta||G_{\beta,\eta}\big) + (\beta+\eta)\log\zeta_{\beta,\eta}, \tag{19}$$

where $\zeta_{\beta,\eta} = \int g_{\beta,\eta}(y)\,dy$. One can see that the above is maximized when $D_{KL}\big(\pi_\theta||G_{\beta,\eta}\big) = 0$, which occurs when $\pi_\theta = G_{\beta,\eta}$. $\qquad\square$

Intuitively, one can see the entropy regularizer $\eta$ as playing the role of "tempering" the reference distribution $\pi_{\text{ref}}$ (larger $\eta$ drives $\pi_{\text{ref}}$ to become more uniform), while both $\beta$ and $\eta$ lower the reward's effect on the target distribution. For the **KL-only case** ($\eta = 0$), the target distribution becomes,

$$G_\beta(y) \propto \pi_{\text{ref}}(y)\exp\left(\frac{R(y)}{\beta}\right), \tag{20}$$

which is the stated result in Remark 3.1. In the **entropy-only case** ($\beta = 0$), the solution is,

$$G_\eta(y) \propto \exp\left(\frac{R(y)}{\eta}\right). \tag{21}$$

All in all, both coefficients play a role in parameterizing the *shape* of the optimal distribution for the regularized RL problem.

## B.2 GRADIENT OF REVERSE-KL REWARD MAXIMIZATION

**Proof of Remark 3.2** From Appendix B.1, we have the identity,

$$-\frac{1}{\beta}J_\beta(\pi_\theta) = D_{KL}\big(\pi_\theta||G_\beta\big) - \log\zeta. \tag{22}$$

We can easily show that the gradient is,

$$\nabla_\theta\left(-\frac{1}{\beta}J_\beta(\pi_\theta)\right) = \nabla_\theta\,D_{KL}\big(\pi_\theta||G_\beta\big) - \nabla_\theta\log\zeta, \tag{23}$$

$$= \nabla_\theta\,D_{KL}\big(\pi_\theta||G_\beta\big). \tag{24}$$

In other words, they are the same up to constant $-\beta$,

$$\nabla_\theta J_\beta(\pi_\theta) = -\beta\,\nabla_\theta D_{KL}\big(\pi_\theta||G_\beta\big). \tag{25}$$

## B.3 TARGET DISTRIBUTION OF FORWARD-KL REWARD MAXIMIZATION

We are interested in finding the distribution $\pi_\theta = G_{\text{fwd}}$ which maximizes,

$$J_{\text{fwd}}(\pi_\theta) = \mathbb{E}_{\pi_\theta(y)}[R(y)] - \beta\,D_{KL}\big(\pi_{\text{ref}}||\pi_\theta\big). \tag{26}$$

Note we can simplify the expression to only terms that depend on $\pi_\theta$,

$$\arg\max_{\pi_\theta} J_{\text{fwd}}(\pi_\theta) = \arg\max_{\pi_\theta}\mathbb{E}_{\pi_\theta}\big[R(y)\big] - \beta\,D_{KL}\big(\pi_{\text{ref}}||\pi_\theta\big), \tag{27}$$

$$= \arg\max_{\pi_\theta}\int\pi_\theta(y)\,R(y) + \beta\,\pi_{\text{ref}}(y)\log\pi_\theta(y)\,dy + \text{const}. \tag{28}$$

**Remark B.1.** *Assuming the reward is finite and has maximum value $R_{max}$. If any on-support answer(s) have $R(y) = R_{max}$, $y \in supp(\pi_{ref})$, the optimal distribution maximizing $J_{fwd}$ will put zero mass outside of $supp(\pi_{ref})$.*

*Proof.* Let $\mathbf{M}$ be the set of on-support, max reward answers: $\mathbf{M}$ if $R(y) = R_{\max}$, and $\mathbf{M} \subseteq supp(\pi_{ref})$. We can generically write any distribution $\pi$ that puts non-zero mass outside of $supp(\pi_{ref})$ as,

$$J_{fwd}(\pi) = C + \int_{\mathbf{M}} \pi(y) R_{\max} + \beta\, \pi_{ref}(y) \log \pi(y)\, dy + \int_{y \notin supp(\pi_{ref})} \pi(y) R(y)\, dy\,, \quad (29)$$

where $C$ captures the contribution to the objective from the remaining $y \in supp(\pi_{ref}), y \notin \mathbf{M}$. Note the forward KL penalty for $y \notin supp(\pi_{ref})$ is zero. We show we can always construct an alternative distribution, $\pi'$, with mass only inside of $supp(\pi_{ref})$ and has strictly higher $J_{fwd}$. We write,

$$J_{fwd}(\pi') = C + \int_{\mathbf{M}} \Big(\pi(y) + \alpha(y)\Big) R_{\max} + \beta\, \pi_{ref}(y) \log\Big(\pi_\theta(y) + \alpha(y)\Big)\, dy\,, \quad (30)$$

where $\alpha$ is a function that redistributes the mass outside of $supp(\pi_{ref})$ across $\mathbf{M}$; $\alpha(y) > 0$, $\int_{y \notin supp(\pi_{ref})} \pi(y)\, dy = \int_{\mathbf{M}} \alpha(y)\, dy$.

First, we note the reward contribution do not decrease from $\pi$ (left hand side) to $\pi'$ (right hand side),

$$\int_{\mathbf{M}} \pi(y) R_{\max}\, dy + \int_{y \notin supp(\pi_{ref})} \pi(y) R(y)\, dy \le \int_{\mathbf{M}} \pi(y) R_{\max}\, dy + \int_{\mathbf{M}} \alpha(y) R_{\max}\, dy\,, \quad (31)$$

since $\int_{y \notin supp(\pi_{ref})} \pi(y)\, dy = \int_{\mathbf{M}} \alpha(y)\, dy$ and $R(y) \le R_{\max}$.

Second, note the (simplified) KL contribution is strictly larger in $\pi'$ (right hand side),

$$\int_{\mathbf{M}} \pi_{ref}(y) \log \pi(y)\, dy < \int_{\mathbf{M}} \pi_{ref}(y) \log\big(\pi(y) + \alpha(y)\big)\, dy\,, \quad (32)$$

since $\log$ is a strictly increasing function, and $\alpha(y) > 0$. Therefore, we have established that,

$$J_{fwd}(\pi) < J_{fwd}(\pi')\,. \quad (33)$$

That is, there always exists a more optimal solution with support solely inside of $supp(\pi_{ref})$. $\qquad\square$

**Proof of Remark 3.3** Assume the reward $R$ is finite and some samples from within $supp(\pi_{ref})$ has $R(y) = R_{\max}$. We optimize with $\beta > 0$ over the restricted feasible set $\Pi$, $\pi_\theta \in \Pi$, where $\pi(y) > 0$ almost everywhere on $supp(\pi_{ref})$ to avoid dividing by zeros.

We write the maximization objective subject to constraints $\int \pi(y)\, dy = 1$, $\pi(y) \ge 0$ for all $y$,

$$\mathcal{L}_J[\pi; \lambda] = \int \pi(y) R(y) + \beta \pi_{ref}(y) \log \pi(y)\, dy + \lambda \Big(\int \pi(y)\, dy - 1\Big) + \int \mu(y)\pi(y)\, dy\,, \quad (34)$$

$$= \int \pi(y) R(y) + \lambda \pi(y) + \beta \pi_{ref}(y) \log \pi(y) + \mu(y)\pi(y)\, dy - \lambda\,, \quad (35)$$

where at the optimal solution, $\mu(y) \ge 0$ and $\mu(y)\pi(y) = 0$.

We take the Gateaux derivative in any perturbation direction $\varphi(y)$, $\int \varphi(y)\, dy = 0$, $\pi(y) + \varepsilon\varphi(y) > 0$,

$$d\,\mathcal{L}_J[\pi; \lambda] = \frac{d}{d\varepsilon} \mathcal{L}_J[\pi + \varepsilon\varphi; \lambda]\Big|_{\varepsilon=0}\,, \quad (36)$$

Defining $0 \log 0 = 0$ per convention. We first solve,

$$\frac{d}{d\varepsilon} \mathcal{L}_J[\pi + \varepsilon\varphi; \lambda] = \frac{d}{d\varepsilon} \int \big(\pi(y) + \varepsilon\varphi(y)\big) R(y) + \lambda\big(\pi(y) + \varepsilon\varphi(y)\big)$$

$$+ \beta\, \pi_{ref}(y) \log\big(\pi(y) + \varepsilon\varphi(y)\big)$$

$$+ \mu(y)\big(\pi(y) + \varepsilon\varphi(y)\big)\, dy\,, \quad (37)$$

$$= \int \varphi(y) R(y) + \lambda\varphi(y) + \beta \frac{\pi_{ref}(y)\, \varphi(y)}{\pi(y) + \varepsilon\varphi(y)} + \mu(y)\varphi(y)\, dy\,, \quad (38)$$

$$= \int \varphi(y) \Big[R(y) + \lambda + \beta \frac{\pi_{ref}(y)}{\pi(y) + \varepsilon\varphi(y)} + \mu(y)\Big] dy\,. \quad (39)$$

$$\frac{d}{d\varepsilon}\mathcal{L}_J[\pi + \varepsilon\varphi; \lambda]\bigg|_{\varepsilon=0} = \int \varphi(y)\left[R(y) + \lambda + \beta\frac{\pi_{\text{ref}}(y)}{\pi(y)} + \mu(y)\right]dy. \tag{40}$$

Define the functional derivative to be,

$$\frac{\delta}{\delta\pi}\mathcal{L}_J[\pi; \lambda] = R(y) + \lambda + \beta\frac{\pi_{\text{ref}}(y)}{\pi(y)} + \mu(y) \tag{41}$$

To find the optimum $\pi^*$ which gives $d/d\varepsilon\,\mathcal{L}_J[\pi + \varepsilon\varphi; \lambda] = 0$ for all $\varphi$, the fundamental lemma of the calculus of variations (Gelfand & Fomin (1963), Lemma 1) tells us it would imply $\delta/\delta\pi\,\mathcal{L}_J[\pi; \lambda] = 0$. Solving for this,

$$R(y) + \lambda + \beta\frac{\pi_{\text{ref}}(y)}{\pi^*(y)} + \mu(y) = 0, \tag{42}$$

$$\Rightarrow \pi^*(y) = \frac{\beta\pi_{\text{ref}}(y)}{-\lambda - R(y) - \mu(y)}, \tag{43}$$

$$\Rightarrow \pi^*(y) = \frac{\beta\pi_{\text{ref}}(y)}{\Lambda - \big(R(y) + \mu(y)\big)}, \qquad \text{define } \Lambda = -\lambda. \tag{44}$$

Per our assumption that some max reward samples are within $\text{supp}(\pi_{\text{ref}})$, Remark B.1 states $\pi^*(y) = 0$ for all $y \notin \text{supp}(\pi_{\text{ref}})$. We can thus ignore the $\pi_{\text{ref}}(y) = 0$ regions. Further observe $\pi_{\text{ref}}(y) > 0$ implies $\pi^*(y) > 0$, thus $\mu(y) = 0$ (per $\pi(y)\mu(y) = 0$). The optimal distribution is therefore,

$$G_{\text{fwd}}(y) = \frac{\beta\pi_{\text{ref}}(y)}{\Lambda - R(y)}, \qquad \Lambda > R_{\max}, \tag{45}$$

where $\Lambda$ is the unique solution to $\int \beta\pi_{\text{ref}}(y)/(\Lambda - R(y))\,dy = 1$. To see this solution exists, observe as $\Lambda \to R_{\max}$, $G_{\text{fwd}}$ at this point goes to infinity. On the other hand, as $\Lambda \to \infty$, all $G_{\text{fwd}}*(y) \to 0$. By continuity, some $\Lambda$ exists between $R_{\max}$ and $\infty$ which satisfy normalization to 1.

Note Grill et al. (2020), Appendix B.3 arrives at a similar solution for the setting of discrete action spaces (i.e. $\pi_\theta$ is a vector).

**When does $G_{\text{fwd}}$ have mass outside of $\text{supp}(\pi_{\text{ref}})$?**    Interestingly, when regularizing with the forward KL, there *are* cases where the optimal distribution $G_{\text{fwd}}$ puts probability density on regions *outside* of the support of $\pi_{\text{ref}}$. First, note that when $\pi_{\text{ref}}(y) = 0$, the KL penalty is zero. We can use this to solve for a simplified version of Equation 42,

$$R(y) + \lambda + \mu(y) = 0, \tag{46}$$
$$\Rightarrow \Lambda = R(y) + \mu(y), \qquad \Lambda = -\lambda. \tag{47}$$

This implies a few possible scenarios for regions where $\pi_{\text{ref}}(y) = 0$,

- If $R(y) < \Lambda$, then $\mu(y) > 0$, implying $\pi(y) = 0$ to respect $\mu(y)\pi(y) = 0$,
- If $R(y) = \Lambda$, then $\mu(y) = 0$, meaning $\pi(y)$ can be positive,
- $R(y) > \Lambda$ is impossible, as $\mu(y) \geq 0$.

Denote $R_{\max}^{\text{in}} = \max_{y\in\text{supp}(\pi_{\text{ref}})} R(y)$ as the on-support max reward, and $R_{\max}^{\text{out}} = \max_{y\notin\text{supp}(\pi_{\text{ref}})} R(y)$ as the off-support max reward. Per Remark B.1, $G_{\text{fwd}}$ will never leave the support of $\pi_{\text{ref}}$ as long as $R_{\max}^{\text{in}} \geq R_{\max}^{\text{out}}$. We therefore consider the case where better samples can be found outside of $\text{supp}(\pi_{\text{ref}})$, $R_{\max}^{\text{out}} > R_{\max}^{\text{in}}$.

Denote an integral over $\pi_{\text{ref}}$ as,

$$Z(c) = \int_{\text{supp}(\pi_{\text{ref}})} \frac{\beta\pi_{\text{ref}}(y)}{c - R(y)}\,dy. \tag{48}$$

Now consider the off-support set with constant max rewards: $\mathbf{M}' = \{y \notin \text{supp}(\pi_{\text{ref}}) : R(y) = R_{\max}^{\text{out}}\}$. Recall this set has higher reward than anything within the support of $\pi_{\text{ref}}$, $R_{\max}^{\text{out}} > R_{\max}^{\text{in}}$. If $Z(R_{\max}^{\text{out}}) < 1$, $\Lambda < R_{\max}^{\text{out}}$ violates impossibility of $R(y) > \Lambda$ above, while $\Lambda > R_{\max}^{\text{out}}$ implies no mass can be placed off support, without normalization on-support ($Z(\Lambda) < 1$). Thus, the only valid solution is $\Lambda = R_{\max}^{\text{out}}$, with the leftover $1 - Z(R_{\max}^{\text{out}})$ probability mass allocated to $\mathbf{M}'$. On the other hand, if $Z(R_{\max}^{\text{out}}) \geq 1$, it implies some $\Lambda \geq R_{\max}^{\text{out}}$ exists which normalizes the on-support distribution and no mass is placed off $\pi_{\text{ref}}$'s support.

## B.4 GRADIENT OF FORWARD-KL REGULARIZED REWARD MAXIMIZATION

**Proof of Remark 3.4** We want to know if optimizing the forward-KL *regularized* RL objective within the support of $\pi_{\text{ref}}$ is equivalent to optimizing a forward KL divergence. In other words, we are interested in whether the following gradient,

$$\nabla_\theta J_{\text{fwd}}(\pi_\theta) = \nabla_\theta \Big[ \mathbb{E}_{\pi_\theta(y)}[R(y)] - \beta\, D_{KL}\big(\pi_{\text{ref}}||\pi_\theta\big) \Big]\,, \tag{49}$$

is a gradient of a forward KL between $\pi_\theta$ and *some* target distribution $h$ that is independent of $\pi_\theta$. We prove by contradiction. Suppose $h$ exists, it follows that the functional derivative of these two objectives must be equivalent up to proportionality,

$$\frac{\delta}{\delta\pi} J_{\text{fwd}}(\pi) \propto \frac{\delta}{\delta\pi} D_{KL}\big(h||\pi\big)\,, \tag{50}$$

where both are subject to constraint $\int \pi(y)\, dy = 1$.

We have established from Equation 41 that the functional derivative of $J_{\text{fwd}}$ subject to constraint $\int \pi(y)\, dy = 1$ is,

$$\frac{\delta}{\delta\pi} \mathcal{L}_J[\pi; \lambda] = R(y) + \beta \frac{\pi_{\text{ref}}(y)}{\pi(y)} + \lambda\,. \tag{51}$$

To find the functional derivative of the forward-KL, we first write down the forward KL objective subject to constraint,

$$\mathcal{L}_K[\pi, \lambda'] = D_{KL}\big(h||\pi\big) + \lambda'\Big(\int \pi(y)\, dy - 1\Big)\,, \tag{52}$$

$$= \int h(y)\log h(y) - h(y)\log \pi(y)\, dy + \int \lambda'\pi(y)\, dy - \lambda'\,, \tag{53}$$

$$= \int \lambda'\,\pi(y) - h(y)\log \pi(y)\, dy + \Big[\int h(y)\log h(y)\, dy - \lambda'\Big]\,, \tag{54}$$

where the right-hand bracket is independent of $\pi$. The Gateaux derivative is,

$$\frac{d}{d\varepsilon}\mathcal{L}_K[\pi + \varepsilon\varphi, \lambda'] = \frac{d}{d\varepsilon}\int \lambda'\big(\pi(y) + \varepsilon\varphi(y)\big) - h(y)\log\big(\pi(y) + \varepsilon\varphi(y)\big)\, dy\,, \tag{55}$$

$$= \int \lambda'\varphi(y) - \frac{h(y)\varphi(y)}{\pi(y) + \varepsilon\varphi(y)}\, dy\,. \tag{56}$$

$$\frac{d}{d\varepsilon}\mathcal{L}_K[\pi + \varepsilon\varphi, \lambda']\Big|_{\varepsilon=0} = \int \varphi(y)\Big[\lambda' - \frac{h(y)}{\pi(y)}\Big]\, dy \tag{57}$$

The functional derivative of the forward KL with respect to the right-hand term is therefore,

$$\frac{\delta}{\delta\pi}\mathcal{L}_K[\pi, \lambda'] = \lambda' - \frac{h(y)}{\pi(y)}\,. \tag{58}$$

Assuming the functional derivative of the two objectives are proportional to each other, we can solve for the target distribution $h(y)$,

$$\frac{\delta}{\delta\pi}\mathcal{L}_K[\pi, \lambda'] \propto \frac{\delta}{\delta\pi}\mathcal{L}_J[\pi, \lambda]\,, \tag{59}$$

$$\Rightarrow \lambda' - \frac{h(y)}{\pi(y)} = \alpha\Big[R(y) + \beta\frac{\pi_{\text{ref}}(y)}{\pi(y)} + \lambda\Big]\,, \qquad \text{for some constant } \alpha\,, \tag{60}$$

$$\Rightarrow h(y) = \big(\lambda' - \alpha\lambda - \alpha R(y)\big)\pi(y) - \alpha\beta\pi_{\text{ref}}(y)\,. \tag{61}$$

Observe one cannot write $h(y)$ independently of $\pi(y)$, other than in trivial cases (e.g. if $R(y)$ is constant such that const $- R(y) = 0$). Thus, for general reward functions $R$, optimizing the forward-KL does not produce a forward KL gradient toward any distribution that can be expressed independently of $\pi_\theta$.

## B.5 Gradient of the forward KL

**Remark B.2.** *The gradient of the forward KL divergence between policy $\pi_\theta$ and target $G_\beta$ is,*

$$\nabla_\theta \, D_{KL}\big(G_\beta||\pi_\theta\big) = -\mathbb{E}_{G_\beta}\big[\nabla_\theta \log \pi_\theta(y)\big]. \tag{62}$$

*Proof.*

$$\nabla_\theta \, D_{KL}\big(G_\beta||\pi_\theta\big) = \nabla_\theta \mathbb{E}_{G_\beta}\big[\log G_\beta(y) - \log \pi_\theta(y)\big], \tag{63}$$

$$= \mathbb{E}_{G_\beta}\big[\nabla_\theta\big(\log G_\beta(y) - \log \pi_\theta(y)\big)\big], \tag{64}$$

$$= -\mathbb{E}_{G_\beta}\big[\nabla_\theta \log \pi_\theta(y)\big]. \tag{65}$$

$\square$

We see that optimizing the forward KL gradient amounts to doing maximum likelihood / supervised fine-tuning on trajectories sampled from the target distribution $G_\beta$, as is also mentioned in some previous works (Agarwal et al., 2024). This is generally intractable as it requires sampling from $G_\beta$. Nevertheless, estimating expectation under a distribution known only up to normalization (i.e. $\mathbb{E}_{G_\beta}[\cdot]$) is well-studied in Monte-Carlo methods (Robert et al., 1999), and it is conceivable that a number of methods there would prove helpful here. For instance, a number of works in diffusion generative modelling proposes to use particle-based samplers, such as sequential Monte Carlo, to approximate the reward-tilted distribution (Wu et al., 2023; Skreta et al., 2025; Singhal et al., 2025).

## B.6 Probability Ratio Under Optimal Target Distribution

**Proof of Proposition 4.1** For any two samples, $y_1$ and $y_2$, their probability ratio under the target distribution is given by,

$$\frac{G_\beta(y_1)}{G_\beta(y_2)} = \frac{g_\beta(y_1)}{\zeta} \frac{\zeta}{g_\beta(y_2)} = \frac{g_\beta(y_1)}{g_\beta(y_2)}, \tag{66}$$

which only require the unnormalized likelihood as the normalization constant $\zeta$ cancel out. Expanding the terms, we can write the log likeilhood ratio in closed form,

$$\log \frac{G_\beta(y_1)}{G_\beta(y_2)} = \log \pi_{\text{ref}}(y_1) \exp\big(\frac{R(y_1)}{\beta}\big) - \log \pi_{\text{ref}}(y_2) \exp\big(\frac{R(y_2)}{\beta}\big), \tag{67}$$

$$= \log \frac{\pi_{\text{ref}}(y_1)}{\pi_{\text{ref}}(y_2)} + \frac{1}{\beta}\Big(R(y_1) - R(y_2)\Big). \tag{68}$$

## B.7 Target distribution after reward augmentation

**Remark B.3.** *Optimizing the reverse-KL regularized RL objective with the augmented reward function $\bar{R}$ yields the following target distribution, which puts uniformly high mass over all samples above reward threshold $R(y) \geq \tau$,*

$$\bar{G}_\beta(y) \propto \begin{cases} \pi_{ref}(y) \exp\left(\frac{R(y)}{\beta}\right) & \text{if } R(y) < \tau, \\ \pi_{ref}(z) \exp\left(\frac{R(z)}{\beta}\right) & \text{if } R(y) \geq \tau. \end{cases} \tag{69}$$

*Proof.* We have established already in Appendix B.1 that the target distribution of reward maximization with reverse KL regularization is,

$$G_\beta(y) \propto \pi_{\text{ref}}(y) \exp\big(\frac{R(y)}{\beta}\big). \tag{70}$$

Plug in the augmented reward function,

$$\bar{R}(y) = \begin{cases} R(y) & \text{if } R(y) < \tau, \\ R(z) + \beta\big(\log \pi_{\text{ref}}(z) - \log \pi_{\text{ref}}(y)\big) & \text{if } R(y) \geq \tau, \end{cases} \tag{71}$$

which gives us the augmented target distribution,

$$\bar{G}_\beta(y) \propto \pi_{\text{ref}}(y) \exp\big(\frac{\bar{R}(y)}{\beta}\big) . \tag{72}$$

In the $R(y) < \tau$ case, $\bar{R}(y) = R(y)$, and there is no change to the (unnormalized) likelihood. In the $R(y) \geq \tau$ case,

$$\log \pi_{\text{ref}}(y) \exp\big(\frac{\bar{R}(y)}{\beta}\big) = \log \pi_{\text{ref}}(y) + \frac{1}{\beta} \bar{R}(y) , \tag{73}$$

$$= \log \pi_{\text{ref}}(y) + \frac{1}{\beta}\Big(R(z) + \beta\big(\log \pi_{\text{ref}}(z) - \log \pi_{\text{ref}}(y)\big)\Big) , \tag{74}$$

$$= \frac{R(z)}{\beta} + \log \pi_{\text{ref}}(y) + \log \pi_{\text{ref}}(z) - \log \pi_{\text{ref}}(y) \tag{75}$$

$$= \frac{R(z)}{\beta} + \log \pi_{\text{ref}}(z) . \tag{76}$$

Therefore we see in the $R(y) \geq \tau$ case we have,

$$\pi_{\text{ref}}(y) \exp\big(\frac{\bar{R}(y)}{\beta}\big) = \pi_{\text{ref}}(z) \exp\big(\frac{R(z)}{\beta}\big) . \tag{77}$$

$\square$

This formally shows the target will have uniformly high density proportional to $\pi_{\text{ref}}(z) \exp(R(z)/\beta)$ for all samples if their original reward $R(y)$ is above threshold $\tau$. If we pick $z$ to be likely under $\pi_{\text{ref}}$, e.g. $z = \arg\max_y \pi_{\text{ref}}(y)$, we can also see these samples will have the highest probabilities in the target distribution.

## B.8 GRADIENT OF REWARD-AUGMENTED OPTIMIZATION

We also note the MARA gradient estimator for an "above threshold" sample $y_i$ (i.e. $R(y_i) \geq \tau$), when using reverse-KL regularization, can be equivalently constructed as using the anchor sample's reference policy probability $\pi_{\text{ref}}(z)$ in lieu of the actual reference probability $\pi_{\text{ref}}(y_i)$ when constructing the KL gradient estimator. To see this precisely, we know the gradient of the expected reward to be,

$$\nabla_\theta \mathbb{E}_{\pi_\theta}\big[R(y)\big] = \mathbb{E}_{\pi_\theta}\big[R(y) \nabla_\theta \log \pi_\theta(y)\big] , \tag{78}$$

and gradient of the reverse-KL regularizer to be,

$$\nabla_\theta D_{KL}\big(\pi_\theta || \pi_{\text{ref}}\big) = \mathbb{E}_{\pi_\theta}\big[(\log \pi_\theta(y) - \log \pi_{\text{ref}}(y))\nabla_\theta \log \pi_\theta(y)\big] . \tag{79}$$

Denote the reward-augmented objective as $\bar{J}_\beta(\pi_\theta) = \bar{R}(y) - \beta D_{KL}(\pi_\theta || \pi_{\text{ref}})$, where $\bar{R}(y) = R(z) + \beta\big(\log \pi_{\text{ref}}(z) - \log \pi_{\text{ref}}(y)\big)$ and $z$ is the "anchor". The gradient estimator of $\bar{K}_i$ for an "above threshold" sample, $y_i, R(y_i) \geq \tau$, can be written as,

$$\bar{K}_i = \Big(\bar{R}(y_i) - \beta \log \frac{\pi_\theta(y_i)}{\pi_{\text{ref}}(y_i)}\Big)\nabla_\theta \log \pi_\theta(y_i) , \tag{80}$$

$$= \Big(R(z) + \beta \log \frac{\pi_{\text{ref}}(z)}{\pi_{\text{ref}}(y_i)} - \beta \log \frac{\pi_\theta(y_i)}{\pi_{\text{ref}}(y_i)}\Big)\nabla_\theta \log \pi_\theta(y_i) , \tag{81}$$

$$= \Big(R(z) - \beta \log \frac{\pi_\theta(y_i)}{\pi_{\text{ref}}(z)}\Big)\nabla_\theta \log \pi_\theta(y_i) . \tag{82}$$

Intuitively, as the anchor is chosen to have high $\pi_{\text{ref}}$, i.e. $\pi_{\text{ref}}(z) > \pi_{\text{ref}}(y_i)$, this can be interpreted as selectively reducing the KL regularization for high-rewarding samples. Mechanistically, this also suggest an alternative implementation which produces the same gradient when using reverse-KL regularization (Algorithm 2).

---

**Algorithm 2** Mode Anchored Reward Augmentation, alternative implementation. The gradient of this algorithm is equivalent to Algorithm 1 when using reverse-KL regularization.

---

1: Given: initial policy $\pi_\theta$, reference distribution $\pi_{\text{ref}}$, reward function $R$, regularization coefficient $\beta$, threshold of good answers $\tau \in \mathbb{R}$, $\tau \leq \max_y R(y)$, and trajectory batch $\{y_i\}_{i=1}^N \sim \pi_\theta$.
2: Pick anchor trajectory: $z = \arg\max_{y_i} \pi_{\text{ref}}(y_i)$, s.t. $R(y_i) \geq \tau$
3: **for** each $y_i$ in batch **do**
4:     **if** $R(y_i) \geq \tau$ **then**
5:         Augment: reward $\bar{r}_i = R(z)$, reference prob $\bar{p}_i = \pi_{\text{ref}}(z)$
6:     **else**
7:         Keep same: reward $\bar{r}_i = R(y_i)$, reference prob $\bar{p}_i = \pi_{\text{ref}}(y_i)$
8:     **end if**
9: **end for**
10: Optimize policy parameters $\theta$ using augmented rewards $\{\bar{r}_i\}_{i=1}^N$ and augmented reference policy probabilities $\{\bar{p}_i\}_{i=1}^N$

---

## C ADDITIONAL EXPERIMENTAL DETAILS

### C.1 DIDACTIC EXPERIMENTS

We construct our didactic experiment as a vector of size 100 (akin to a output space with 100 tokens). We initialize a categorical distribution over this output space whose logits are all 0's (i.e. uniform distribution over all tokens). Given some reward function and reference distribution defined over this space, we optimize this categorical distribution with the KL-regularized policy gradient for 3000 gradient steps in PyTorch with Adam optimizer, with learning rate 5e-3 and batch size 32.

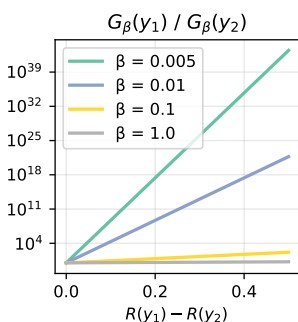

Figure 7: Effect of reward difference ($\Delta R$, x-axis) and reverse-KL regularization strength (hue) on the relative probabilities between two samples in the optimal policy distribution (y-axis)

### C.2 THE 1-2 TASK

We ask the LM to generate a uniform random integer that is either 1 or 2 (Hopkins et al., 2023), as illustrated in Figure 9. We run for a range of KL coefficients ($\beta$) and multiple random seeds. Figure 10 shows the training run for just vanilla RL, without MARA.

### C.3 CREATIVE QUESTION ANSWERING TASK

We detail the training settings in Table 3, and evaluation settings in Table 4. We follow the evaluation procedures outlined in both Kirk et al. (2023) and Zhang et al. (2025). The specific evaluation metrics are defined as follows.

- `In Dist Reward`: training reward, on training set, using training reward model
- `Out Dist Reward`: evaluation reward on held-out set, using evaluation reward model
- `Ngram EAD`: Expectation-adjusted Distinct N-gram, proposed in Liu et al. (2022). We follow (Kirk et al., 2023) and average EAD for $n = 1, ..., 5$

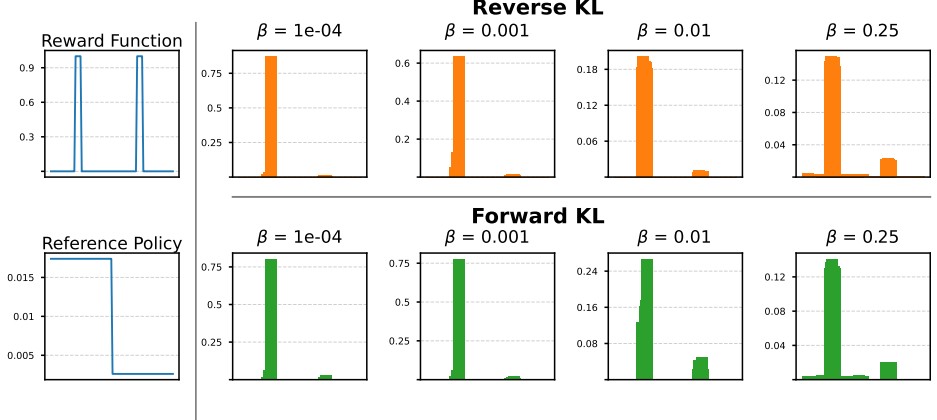

Figure 8: Final policy distribution after KL-regularized RL, with equal rewards for all correct answers. Low-support (yet equally correct) answers are never preferred over high-support answers.

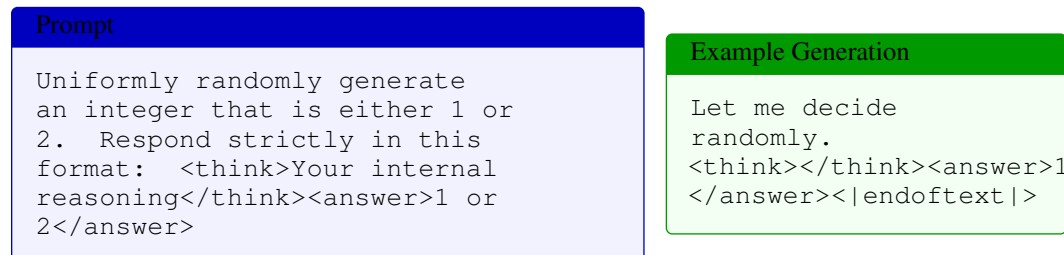

Figure 9: The 1-2 task to test output distribution of LMs.

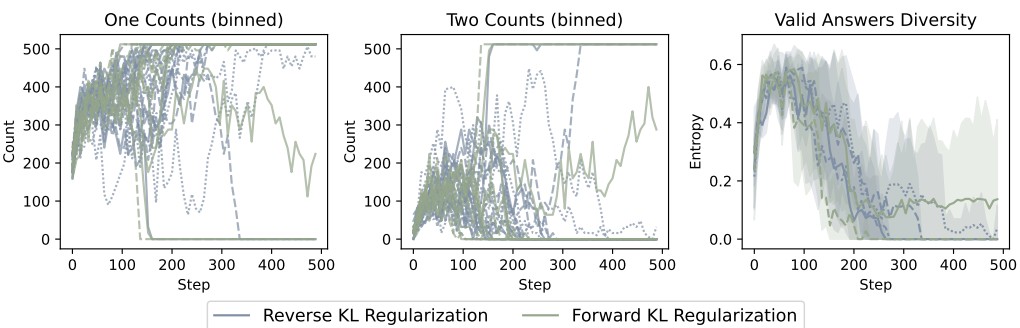

Figure 10: Training outcomes using vanilla RL. **(Left, Middle)** Policy's empirical distribution over valid answers for runs that reached high rewards (counts binned over 8 consecutive training batches), across a range of regularization coefficients ($\beta$). **Right** Diversity of the valid answers over the course of training, measured as the entropy of the Bernoulli distribution over answers of 1's and 2's.

| Hyperparameter | Value |
|---|---|
| Actor Model | `Qwen3-1.7B` |
| Reward Model | `Skywork-Reward-V2-Qwen3-4B` |
| Training Dataset | `Wildchat 10k English` |
| Train Batch Size | 128 |
| Mini-Batch Size | 64 |
| Max Prompt Length | 512 |
| Max Response Length | 2048 |
| Learning Rate | $1 \times 10^{-6}$ |
| Entropy Coefficient | 0 |
| Rollout n (per prompt) | 5 |
| Gradient Checkpointing | Enabled |
| Epochs | 3 |

Table 3: Creative QA Training Setting

| Hyperparameter | Value |
|---|---|
| Evaluation Reward Model | `Skywork-Reward-Gemma-2-27B-v0.2` |
| Dataset | `NoveltyBench curated` |
| Num Generations / Prompt | 10 |
| Max Tokens | 4000 |
| Temperature | 1.0 |
| Enable Thinking (qwen) | `False` |

Table 4: Creative QA Evaluation Setting

- `Semantic Div`: semantic embedding diversity as measured by averaged cosine distance, using embedding model `all-MiniLM-L6-v2`.

- `Mean Distinct`: Estimates a notion of "# of distinct concepts", as introduced in Zhang et al. (2025).

We run additional baselines for the effect of the batch level threshold to set $\tau$ in Table 5.

### C.4 EVIDENCE FOR UNIMODAL TARGET DISTRIBUTIONS IN LMS

We show additional evidence in existing LLMs settings, the shape of the reference distribution and reward function leads to highly skewed target distributions.

First, we draw 8192 samples from `Qwen2.5-3B` using prompt for the 1-2 task (Appendix C.2). We filter for correct answers ($R(y) = 1$), leaving 2944 samples (35.9% correct). We see in Figure 11a the distribution of "1" and "2", with 1 being over-represented, pointing to a skew in the base reference distribution favouring "1", despite the model being prompted to uniformly randomly generate an integer.

| Model | Out-dist Reward (↑) | EAD (↑) | Semantic Div (↑) | Distinct (↑) |
|---|---|---|---|---|
| MARA (rev, 0.90) | 1.451 ±0.103 | 0.543 ±0.014 | 0.186 ±0.008 | 4.14 ±0.233 |
| MARA (fwd, 0.90) | 1.604 ±0.113 | 0.568 ±0.012 | 0.193 ±0.009 | 4.62 ±0.258 |
| MARA (rev, 0.75) | 1.498 ±0.117 | 0.547 ±0.013 | 0.183 ±0.008 | 4.41 ±0.262 |
| MARA (fwd, 0.75) | 1.325 ±0.097 | 0.508 ±0.014 | 0.196 ±0.009 | 4.07 ±0.243 |

Table 5: Ablation of batch-level threshold for $\tau$, set at either 90th percentile (0.90) or 70th percentile (0.75). Mean ± bootstrap SEM.

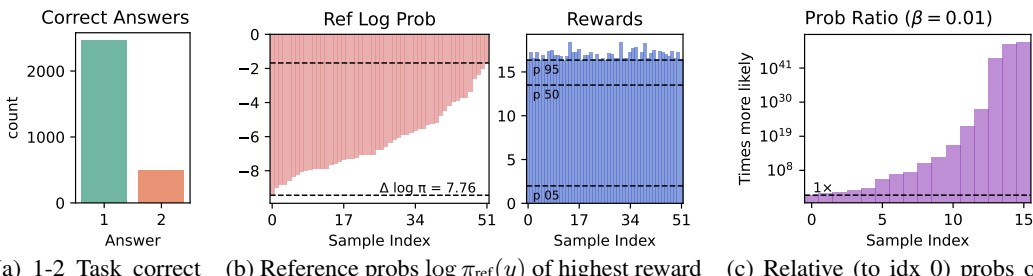

(a) 1-2 Task correct answers distribution

(b) Reference probs $\log \pi_{\text{ref}}(y)$ of highest reward samples in Creative QA task; $\Delta \log \pi$ shows range

(c) Relative (to idx 0) probs of most likely samples under $G_\beta$

Figure 11: Evidence for highly skewed target distributions $G_\beta$ in LLM tasks

For creative QA (Appendix C.3), we draw 1024 samples from `Qwen3-1.7B` using a single question in WildChat text. We evaluate the responses using `Skywork-Reward-V2-Qwen3-4B` (reward mean: 12.33, min: $-5.125$, max: 18.38), and filter for answers above the 95th percentile ($R = 16.37$). This leaves 52 samples (reward mean: 17.0, min: 16.38, max: 18.38). We see in Figure 11b that among these high-reward answers, $\log \pi_{\text{ref}}(y)$ has a difference of up to 7.76, corresponding to a probability ratio of $2345\times$ under $G_\beta$ (Remark 4.3).

We then estimate the probability ratio taking into account both the reference probability and the reward (Proposition 4.1). We use the same 1024 samples and compute their unnormalized likelihood $G_\beta$, $\pi_{\text{ref}}(y) \exp(R(y)/\beta)$, with $\beta = 0.01$. We take the top 16 most likely samples and calculate their relative probabilities *with respect to the 16th most likely sample*, such that the lowest probability ratio is $1\times$. We observe in Figure 11c that amongst the sampled responses, the most probable sample ($R = 18.4$, $\log \pi_{\text{ref}}(y) = -5.91$) is $2 \times 10^{49}$ times more likely under the target distribution $G_\beta$ than the 16th most likely sample ($R = 17.3$, $\log \pi_{\text{ref}}(y) = -7.09$), despite having only slightly higher rewards and reference probabilities.

## C.5 DRUG DISCOVERY

Chemical language models (CLMs) that generate molecules in string-based formats, e.g., a SMILES string (Weininger, 1988), have been experimentally validated with numerous generated molecules in clinical trials (Du et al., 2024). Recently, the field has focused on addressing "synthesizability", i.e., can generated molecules actually be synthesized in the lab? (Stanley & Segler, 2023; Papidocha et al., 2025). Accordingly, we adapt two reward functions from Guo et al. (2025b): `SYNTH` and `SYNTH-ALL-AMIDE` that jointly reward binding potency and synthesizability. `REINVENT` (Olivecrona et al., 2017) is a state-of-the-art RL-based CLM on standard benchmarks (Gao et al., 2022). The recent Saturn CLM (Guo & Schwaller, 2024b) notably improves optimization efficiency by using data augmentation (Bjerrum, 2017; Guo & Schwaller, 2024a), but continues to use `REINVENT`'s RL algorithm.

In the drug discovery experiments adapted from Guo et al. (2025b), the reward functions are comprised of numerous individual optimization objectives, and defines a multi-parameter optimization task. Concretely, these objectives are:

1. *Minimize* the molecular docking score using QuickVina2-GPU (Trott & Olson, 2010; Alhossary et al., 2015; Tang et al., 2024). Docking simulates binding of molecules to a target protein and predicts a crude binding affinity value. Docking was performed against the ATP-dependent Clp protease proteolytic subunit (ClpP) Mabanglo et al. (2023).

2. *Maximize* the quantitative estimate of drug-likeness (QED) (Bickerton et al., 2012), which is itself comprised of various physico-chemical properties, e.g., molecular weight. Maximizing QED can prevent generated molecules from being too large and lipophilic.

3. *Constrain* the number of hydrogen-bond donors (HBDs): HBDs $< 4$. This can improve absorption, Distribution, metabolism, and excretion (ADME) properties (Kenny, 2022) of the generated molecules.

4. *Satisfy* the "Synthesizability" constraint. Synthesizability is quantified by using a retrosynthesis model on each generated molecule. Retrosynthesis models predict a plausible

synthesis route to synthesize a target molecule using commercially available precursors. The precursors set is from the eMolecules catalogue extracted from Chen et al. (2020). Retrosynthesis models typically start with a "single-step" model which predicts precursors given a target molecule. Since molecules may require multiple steps to synthesize, "Multi-step Retrosynthesis" commonly couples a search algorithm with single-step models to iteratively decompose a target molecule. In this work, we use the MEGAN (Sacha et al., 2021) single-step model with the Retro* (Chen et al., 2020) search algorithm using the Syntheseus (Maziarz et al., 2025) package. Finally, a molecule is considered synthesizable if the retrosynthesis model successfully proposes a synthesis route.

Both the SYNTH and SYNTH-ALL-AMIDE reward functions are comprised of the above objectives. The only difference is that in the SYNTH-ALL-AMIDE case, a molecule is *only* considered synthesizable if all the chemical reactions involved to synthesize it are "amide coupling reactions". Amide couplings are one of the most common reactions performed in the pharmaceutical industry (Brown & Bostrom, 2016), and is generally a robust, widely compatible reaction. Subsequently, the reward function is defined as a product of each individual component above. Given a molecule, $x$:

$$R(x) = DS(x) \times QED(x) \times HBD(x) \times Syntheseus(x) \in [0, 1] \tag{83}$$

where "DS" is docking score. The HBD and Syntheseus objectives are binary, i.e., 1 if satisfied and 0 otherwise. $QED \in [0, 1]$ and is used as is. The QuickVina2-GPU docking score is reward shaped using a reverse sigmoid function following Guo et al. (2025b) and gives higher reward to lower docking scores, as desired.

Our goal in this section is to investigate the potential for MARA to be a *drop-in replacement* for the REINVENT (Olivecrona et al., 2017) RL-based algorithm for molecular design. REINVENT is amongst the most performant molecular design algorithms (Gao et al., 2022) and the Saturn model (Guo & Schwaller, 2024b) adapts this algorithm and leverages data augmentation (Bjerrum, 2017; Guo & Schwaller, 2024a) to further improve optimization efficiency.

We evaluate all models with a fixed budget of 10,000 reward function evaluations, which is standard in benchmarks. We contrast the algorithms' performance on molecular design metrics that measure optimization efficiency and diversity. Yield is the number of *unique* molecules above a reward threshold. OB100 is the number of reward evaluations required to generate 100 molecules above the same threshold. IntDiv1 (Polykovskiy et al., 2020) and #Circles (Xie et al., 2023) are diversity metrics based on molecular sub-structure based features, and measure intra-set similarity and sphere packing, respectively.

Tables 6 and 7 show the optimization results for the SYNTH and SYNTH-ALL-AMIDE reward functions at the 0.80 and 0.85 screening thresholds, respectively. MARA is trained with $\tau = 0.80$ in both. In general, MARA matches or outperforms REINVENT particularly for the more challenging SYNTH-ALL-AMIDE reward function. In this environment, MARA can find more high-reward molecules (Yield) and using less reward evaluations (OB100) than REINVENT.

Table 6: Results at Threshold $= 0.8$ ($\uparrow$ larger is better; $\downarrow$ smaller is better). "SYNTH" and "AMIDE" denote the `SYNTH` and `SYNTH-ALL-AMIDE` reward functions, respectively.

| Task | Algorithm | Sigma | Gen Yield ($\uparrow$) | OB100 ($\downarrow$) | IntDiv1 ($\uparrow$) | Circles ($\uparrow$) |
|------|-----------|-------|-----------|-------|---------|---------|
| SYNTH | REINVENT | 128 | $6569 \pm 186$ | $1042 \pm 66$ | $0.766 \pm 0.011$ | $67 \pm 3$ |
| | | 256 | $6618 \pm 93$ | $1080 \pm 89$ | $0.756 \pm 0.012$ | $57 \pm 8$ |
| | | 512 | $6746 \pm 161$ | $1067 \pm 74$ | $0.752 \pm 0.016$ | $55 \pm 5$ |
| | MARA | 128 | $6834 \pm 78$ | $1015 \pm 55$ | $0.761 \pm 0.009$ | $59 \pm 8$ |
| | | 256 | $6750 \pm 139$ | $1068 \pm 50$ | $0.760 \pm 0.012$ | $60 \pm 4$ |
| | | 512 | $6793 \pm 267$ | $1065 \pm 49$ | $0.751 \pm 0.015$ | $60 \pm 1$ |
| AMIDE | REINVENT | 128 | $5433 \pm 184$ | $1427 \pm 63$ | $0.768 \pm 0.012$ | $35 \pm 1$ |
| | | 256 | $5544 \pm 172$ | $1406 \pm 59$ | $0.768 \pm 0.009$ | $34 \pm 5$ |
| | | 512 | $5334 \pm 165$ | $1445 \pm 111$ | $0.776 \pm 0.008$ | $33 \pm 4$ |
| | MARA | 128 | $5635 \pm 249$ | $1407 \pm 123$ | $0.766 \pm 0.008$ | $36 \pm 3$ |
| | | 256 | $5353 \pm 114$ | $1393 \pm 42$ | $0.769 \pm 0.009$ | $33 \pm 4$ |
| | | 512 | $5377 \pm 152$ | $1343 \pm 77$ | $0.763 \pm 0.008$ | $31 \pm 3$ |

Table 7: Results at Threshold $= 0.85$ ($\uparrow$ larger is better; $\downarrow$ smaller is better). "SYNTH" and "AMIDE" denote the `SYNTH` and `SYNTH-ALL-AMIDE` reward functions, respectively.

| Task | Algorithm | Sigma | Gen Yield ($\uparrow$) | OB100 ($\downarrow$) | IntDiv1 ($\uparrow$) | Circles ($\uparrow$) |
|------|-----------|-------|-----------|-------|---------|---------|
| SYNTH | REINVENT | 128 | $1614 \pm 407$ | $4114 \pm 109$ | $0.701 \pm 0.018$ | $7 \pm 1$ |
| | | 256 | $1552 \pm 242$ | $3940 \pm 371$ | $0.699 \pm 0.030$ | $6 \pm 1$ |
| | | 512 | $1484 \pm 45$ | $3717 \pm 201$ | $0.701 \pm 0.026$ | $6 \pm 1$ |
| | MARA | 128 | $1796 \pm 210$ | $3654 \pm 272$ | $0.716 \pm 0.015$ | $6 \pm 1$ |
| | | 256 | $1530 \pm 126$ | $3957 \pm 335$ | $0.705 \pm 0.014$ | $8 \pm 1$ |
| | | 512 | $1550 \pm 347$ | $4016 \pm 234$ | $0.689 \pm 0.024$ | $6 \pm 1$ |
| AMIDE | REINVENT | 128 | $1098 \pm 88$ | $4360 \pm 257$ | $0.721 \pm 0.016$ | $8 \pm 1$ |
| | | 256 | $1488 \pm 280$ | $4290 \pm 141$ | $0.725 \pm 0.021$ | $8 \pm 1$ |
| | | 512 | $1054 \pm 152$ | $4620 \pm 438$ | $0.739 \pm 0.009$ | $8 \pm 0$ |
| | MARA | 128 | $1235 \pm 130$ | $3943 \pm 303$ | $0.733 \pm 0.009$ | $8 \pm 1$ |
| | | 256 | $1404 \pm 261$ | $4079 \pm 172$ | $0.730 \pm 0.010$ | $7 \pm 1$ |
| | | 512 | $1341 \pm 86$ | $3930 \pm 400$ | $0.723 \pm 0.004$ | $7 \pm 1$ |

