# OpenReview forum: "KL-Regularized Reinforcement Learning for Generative Modelling is Designed to Mode Collapse"
_ICLR.cc/2026/Conference — ICLR 2026 Poster_

### Official Review · Reviewer_K4Yn · 2025-10-21

**Soundness:** 2
**Presentation:** 3
**Contribution:** 3
**Rating:** 4
**Confidence:** 3

**Summary:**

This paper analyzes the reasons for "mode collapse" when using KL-regularized RL for LLM post-training, linking it fundamentally to the shape of the target distribution. For this, the paper revisits earlier-made arguments about the shape of the target distribution that Reverse KL-regularized RL approximates and makes (new?) arguments about the target distribution of Forward KL-regularization. Using insights from this analysis, the authors propose a new MARA algorithm that is claimed to approximate a target distribution with uniform high probability modes for highly-rewarded samples, and sticks to the reference distribution otherwise.

**Strengths:**

- S1) The paper addresses a highly debated topic at the moment on which a lot of noise and confusion exists.
- S2) The paper is quite clearly written and the explanations are easily followable.
- S3) The paper introduces a new method based on a theoretically clear argument.
- S4) The analysis of the forward KL-regularized objective is quite interesting if new, even though I'm not sure it has much impact on the paper's main narrative.

**Weaknesses:**

- W1) I feel that part of the discussion increases --rather than dispelling--  the confusion.
In particular, starting from the first two sentences of the abstract:
  "It is commonly believed that optimizing the reverse KL divergence result in “mode
seeking”, while optimizing forward KL result in “mass covering”, with the latter
being preferred if the goal is to sample from multiple diverse modes. We show—
mathematically and empirically—that this intuition does not necessarily transfer
well to doing reinforcement learning with reverse/forward KL regularization (e.g.
as commonly used with language models)."
I think this is either false or misleading at best. One thing is the kind of regularization used in the loss and another is the divergence implicitly optimized by the training objective (remark 3.2). It is the latter that can be ascribed "mode-seeking" or "mode-covering" behavior, whereas here the authors are referring to the divergence used in the regularization term. If the goal of the paper is clarifying these aspects of LLM post-training using RL, it could do a much better job at keeping this distinction clear.
- W2) Furthermore, there are a number of inaccuracies sprinkled across the paper. One of the most important ones is in one of the main messages of the paper, which is that the shape of the target distribution is responsible for the lack of diversity (again, in the abstract, "Further, we show commonly used settings such as low regularization strength and equal verifiable rewards tend to specify uni-modal target distributions, meaning the optimization objective is by construction non-diverse."). This looks like a misconceived conclusion from the analysis in the paper. The illustration in Section 3.3 corresponds to a reward function that assigns non-uniform values to rewarded samples, whereas the illustration in Figure 5 compares "on-support" with "off-support" sequences (see also Q6 about this terminology). Actually, as noted in Remark 4.3 "[the] relative probabilities in the solution is simply the relative probabilities in the reference
distribution, and do not depend on the KL-regularization strength $\beta$". Doesn't then this contradict the sentence in the abstract?
- W3) While the proposed algorithm is interesting, there is no comparison with other diversity-enhancing baselines in the empirical part such as rewarding the unlikely or pass@k training.

**Questions:**

- Q1) L31 why do you say RL is the only way to train models where the solution is not known a priory? You already cite Go et al. (2023). Wouldn't this fit this bill?
-Q2) What is the role of remarks 3.2, 3.3 and 3.4 in the overall story of the paper? I felt that they were made and forgotten.
-Q3) L174: when you say that ML training on samples from $G_\beta$ is intractable, you mean for an unbounded reward function, right? You mention STaR, but you could make it more explicit that this algorithm is doing exactly what you are saying is intractable for a particular class of rewards on which we can do rejection sampling. More importantly, note that you don't necessarily need to sample from $G_\beta$ to approximate this objective. $G_\beta$ is particular case of target distributions described as "distributional constraints" in https://arxiv.org/abs/2012.11635, and the forward KL objective can be approximated using the DPG algorithm.
- Q4) What is the policy you are using for Fig. 2 and following?
- Q5) Definition 3.5: just to double check, do you really mean that _all_ high-reward samples must have high probability? OK, but this doesn't match my intuition for multimodality. Maybe you are looking for a different concept?
- Q6) When you say "off-support", you mean $p(y)<\epsilon$, right? Correct me if I'm wrong, but I think mathematically off-support means $p(y)=0$. You might need to define this other concept of support to use it here.
- Q7) Many recent papers set $\beta=0$ (e.g. Dr. GRPO). What happens when your algorithm for this particular case in the scenario of verifiable rewards? It looks to me that it boils down to regular RL, and cannot prevent the loss in diversity.
- Q8) L361 "stays close to the reference in regions of low reward" -> why is that the case?
- Q9) Following up, can you say something about the target distribution of MARA?
- Q10) Figure7: the KL parameter in the legend is $\beta$?
- Q11) What is In dist and out dist reward?
- Q12) Could you add a few words about the REINVENT algorithm?
- Q13) Why does the threshold affects REINVENT?
- Q14) "In this work, we provide an in-depth understanding of the KL-regularized RL objective, particularly in terms of its diversity" -> could you summarize it in a few words?

Typos/style/corrections
- L107 "maximize a reward function" -> "maximize the expected value of a reward function"?
- L149 "we can regluarized" -> we can regularize
- L149 "maximizaztion" -> "maximization"
- L208 "token space" -> sequence/output space ??
- Figure 7 caption: "entrpy" -> entropy
- L426 "adpat" -> adapt

---

> ### Author Response · Authors · 2025-11-24
>
> We thank the reviewer for the insightful comments. We are happy to hear the paper is clearly written and addresses a highly debated current topic,  introduces a novel and theoretically clear method, and makes a theoretical contribution for the forward KL regularized case. Indeed, the forward KL analysis is new to our knowledge, though we are happy to add any missed citations to previous works.
>
> We are particularly grateful for the very constructive comments on cleaning up some loose ends in the writing. We have fixed all of them in an updated pdf of the paper. Please see below for detailed responses to each.
>
>
> ## Weaknesses
>
> > W1) Confusing discussion about regularization used in the loss vs. the divergence implicitly optimized by the training objective (remark 3.2). It is the latter that can be ascribed "mode-seeking" or "mode-covering" behavior, whereas here the authors are referring to the divergence used in the regularization term.
>
> We are happy to make the writing clearer. Our intent is in agreement with the reviewer’s comment: to clarify *KL-minimization* and *KL-regularized (reward) optimization* are different things, and that diversity collapse behaviour in the latter depends on more factors than the choice of regularization divergence alone. In our experience, this has been a point of confusion in the current RL for generative modelling community (even if it is perhaps an obvious point for the reviewer).
>
> We have updated our introduction section to:
> - Clearly distinguish *KL-minimization* and *KL-regularized (reward) optimization* as two concepts (as the reviewer have stated above)
> - Motivate studying the *implicitly optimized divergence* between the policy and the target distribution as a more meaningful direction for future research, rather than the regularization divergence
> - (As we have done already) Point out the lack of diversity in the optimal target distribution, which is an issue irrespective of the implicit divergence being optimized
> - State clearly that our working definition of “mode coverage” refers to mode in the reward function, per Def 3.5 (i.e. having a diverse set of good answers)
>
> We welcome any additional notes from the reviewer if they feel it will help further dispel the confusion.
>
>
> > W2) Furthermore, there are a number of inaccuracies… in one of the main messages of the paper, which is that the shape of the target distribution is responsible for the lack of diversity. This looks like a misconceived conclusion from the analysis in the paper. The illustration in Section 3.3 corresponds to a reward function that assigns non-uniform values to rewarded samples, whereas the illustration in Figure 5 compares "on-support" with "off-support" sequences.
>
> The two sections outline two *different*, yet commonly used settings where diversity loss occurs in the target distribution. We have also updated our Figures 4 and 5 to improve clarity, which we attach as image links below for now.
>
> - With unequal reward, equal $\pi_{ref}$, small linear differences in reward lead to large exponential differences in the target relative probabilities, which results in unimodal target solutions around the highest reward solution. This setting is common for continuous reward functions, as seen in LM alignment, and drug discovery. [updated figure](https://ibb.co/dw4jZSmq)
> - With equal reward, unequal $\pi_{ref}$, RL does not change relative probabilities from $\pi_{ref}$. This means if there are samples with vastly different probabilities under $\pi_{ref}$ to begin with, the higher-support sample will still dominate the resulting distribution. [updated figure](https://ibb.co/DHTVsJtz)
>
> We hope this clarifies. The new figures have been added to the paper.

---

> > ### Author Response · Authors · 2025-11-24
> >
> > > W3) While the proposed algorithm is interesting, there is no comparison with other diversity-enhancing baselines in the empirical part such as rewarding the unlikely or pass@k training.
> >
> > We added additional diversity-promoting baselines, including entropy regularization, rewarding the unlikely [1], inference-aware Best-of-N training [2], and weight ensembling [3]. We found the main conclusion remains the same – MARA remains the most performant in 3/4 metrics.
> >
> >
> > | Model              | In-dist Reward (↑) | Out-dist Reward (↑)           | Ngrams EAD (↑)              | SemDiv (↑)            | Mean Distinct  (↑)         |
> > |--------------------|---------|---------------------|------------------|--------------------|--------------------|
> > | Base Model         | 10.94   | 1.166 ± 0.076       | 0.413 ± 0.015     | 0.220 ± 0.009      | 4.01 ± 0.254       |
> > | GRPO               | 14.8    | 1.317 ± 0.102       | 0.497 ± 0.014     | 0.193 ± 0.009      | 3.96 ± 0.249       |
> > | RLOO               | 15.56   | 1.280 ± 0.100       | 0.514 ± 0.014     | 0.192 ± 0.008      | 3.88 ± 0.243       |
> > | Entropy            | 1.44    | 0.786 ± 0.073       | 0.267 ± 0.009     | **0.228 ± 0.009**  | 3.45 ± 0.193       |
> > | Unlikely       | 10.04   | 1.381 ± 0.114       | 0.532 ± 0.015     | 0.191 ± 0.008      | 4.24 ± 0.239       |
> > | BoN training    | **16.88** | 0.596 ± 0.055     | 0.541 ± 0.010     | 0.162 ± 0.008      | 2.29 ± 0.173       |
> > | Ensembling          | -       | 1.143 ± 0.086       | 0.438 ± 0.014     | 0.211 ± 0.010      | 4.19 ± 0.269       |
> > | MARA (rev)         | 15.42   | 1.451 ± 0.103       | 0.543 ± 0.014     | 0.186 ± 0.008      | 4.14 ± 0.233       |
> > | MARA (fwd)         | 15.33   | **1.604 ± 0.113**   | **0.568 ± 0.012** | 0.193 ± 0.009      | **4.62 ± 0.258**   |
> >
> >
> >
> >
> > ## Questions
> >
> > > Q1) L31 why do you say RL is the only way to train models where the solution is not known a priory?
> >
> > We updated this sentence to “RL is the primary way to train models in settings where the correct solution is not known a priori” in the updated paper.
> >
> > > Q2) Point of remarks 3.2, 3.3 and 3.4 in the overall story of the paper? I felt that they were made and forgotten.
> >
> > The point of remarks 3.1-3.4 is to provide general clarity around (i) the optimal solution, and (ii) the gradient of the reverse- and forward-KL regularized RL objectives. While section 4 focuses on the more common reverse KL case, we felt it was more complete to include a full analysis of both KL regularizers (in the spirit of dispelling confusion in the community). We’ve updated our writing in the section for greater clarity.
> >
> >
> > > Q3) L174: when you say that ML training on samples from $G_\beta$ is intractable, you mean for an unbounded reward function? You mention STaR, but you could make it more explicit that this algorithm is doing exactly what you are saying is intractable for a particular class of rewards on which we can do rejection sampling. More importantly, note that you don't necessarily need to sample from $G_\beta$ to approximate this objective. $G_\beta$ is a particular case of target distributions described as "distributional constraints" in [3], and the forward KL objective can be approximated using the DPG algorithm.
> >
> > Many thanks for the input and the additional references! We have made the statement in L174 more precise in the updated paper. Specifically:
> > - Optimizing an unbiased forward KL gradient toward a generic distribution is intractable
> > - In cases where reward is bounded and we know $G_\beta$ up to normalization, filtering can be viewed as rejection sampling to get samples from $G_\beta$
> > - Alternative methods include [4, 5] which approximately sample from $G_\beta$ (will add citation).
> >
> > Let us know if you have any other thoughts / corrections.
> >
> >
> > > Q4) What is the policy you are using for Fig. 2 and following?
> >
> > Fig 2 - 6 use a 100-category categorical distribution. We directly plot each category’s probability for didactic visualization.
> >
> > > Q5) Def 3.5. Do you really mean that all high-reward samples must have high probability? This doesn't match my intuition for multimodality.
> >
> > Yes, within this paper we define multimodality as putting mass over all high-reward samples. We think this is an intuitive working definition for diversity many in the community shares (i.e. want diverse but high quality samples). We are happy to include a discussion of other definitions / intuitions of multimodality if the reviewer has specific ones in mind.
> >
> > > Q6) When you say "off-support", you mean $\pi_{\text{ref}}(y) < \epsilon$, right?
> >
> > Indeed. Apologize for the colloquialism. We have updated all instances of “off support” to “low support” in the updated pdf.

---

> > > ### Author Response · Authors · 2025-11-24
> > >
> > > > Q7) Many recent papers set $\beta = 0$. What happens to MARA when $\beta = 0$? It looks like it cannot prevent the loss in diversity.
> > >
> > > We think the reviewer is correct here. We specifically deal with the regularized RL setting in this paper. When $\beta = 0$ the target distribution is no longer well-defined, and we do not expect our algorithm to work there to prevent diversity collapse. Works such as [6] attempt to preserve distributional properties in the limit of no regularization; we are happy to include a discussion if the reviewer finds it useful.
> > >
> > > > Q8, Q9) L361 "stays close to the reference in regions of low reward" -> why is that the case? What is the target distribution of MARA?
> > >
> > > Indeed. Appendix Remark B.2 states the target distribution of MARA. In short: above the threshold $\tau$, the target puts uniform mass at unnormalized density $\pi_{ref} (z) \exp ( R(z) / \beta )$ . Below the threshold it reduces to the usual reward-tilted reference distribution $\pi_{ref} (y) \exp ( R(y) / \beta )$.
> > >
> > > > Q10) Fig 7: the KL parameter in the legend is $\beta$
> > >
> > > Yes
> > >
> > > > Q11) What is in dist and out dist reward (Table 1)?
> > >
> > > We train with `Skywork-Reward-V2-Qwen3-4B` reward model, and evaluate with `Skywork/Skywork-Reward-Gemma-2-27B-v0.2`. This is to examine generalization ability and reward hacking.
> > >
> > > > Q12) Could you add a few words about the REINVENT algorithm?
> > >
> > > At its core, REINVENT [7] optimizes the loss function $$- [ \log \pi_\theta (y) - (\log \pi_{\text{ref}} (y) + \sigma R(y))]^2$$ which is equivalent to a KL-regularized reward maximization problem.This objective similarly has the mode collapse issues we outline in the paper, and we found that augmenting its core loss with MARA improves performance. We have added the above explanation to the paper.
> > >
> > >
> > > > Q13) Why does the threshold affect REINVENT?
> > >
> > > The threshold (in Table 2) is the **screening threshold** (follows convention of [8]). Higher screening thresholds is a stricter setting and results in lower performance for both models. We have changed the Table 2 column name to “Screen Threshold” to clear up confusion.
> > >
> > > Note MARA is trained with $\tau = 0.8$ for all screen thresholds in the main text. We run an additional experiment where we train MARA with $\tau = 0.85$. This pushes up the Yield even further when Screening Threshold is more strict.
> > >
> > > ### SYNTH task (Table 2a)
> > >
> > > | Screen Thresh | Algorithm             | Yield (↑)   | OB100 (↓)  | IntDiv1 (↑)    | Circles (↑) |
> > > |-------------|------------------------|-------------|------------|----------------|-------------|
> > > | 0.8         | REINVENT               | 6569±186    | 1042±66    | 0.766±0.011    | 67±3        |
> > > | 0.8         | MARA ($\tau = 0.80 $ )      | 6834±78     | 1015±55    | 0.761±0.009    | 59±8        |
> > > | 0.8         | MARA ($ \tau = 0.85 $ )      | 6584±231    | 1042±66    | 0.761±0.008    | 72±6        |
> > > ||||
> > > | 0.85        | REINVENT               | 1614±407    | 4114±109   | 0.701±0.018    | 7±1         |
> > > | 0.85        | MARA ($ \tau = 0.80 $ )      | 1796±210    | 3654±272   | 0.716±0.015    | 6±1         |
> > > | 0.85        | MARA ($ \tau = 0.85$ )      | 2196±394    | 4010±297   | 0.703±0.011    | 7±1         |
> > >
> > >
> > >
> > > ### ALL-AMIDE task (Table 2b)
> > >
> > > | Eval Thresh | Algorithm             | Yield (↑)   | OB100 (↓)  | IntDiv1 (↑)    | Circles (↑) |
> > > |-------------|------------------------|-------------|------------|----------------|-------------|
> > > | 0.8         | REINVENT               | 5433±184    | 1427±63    | 0.768±0.012    | 35±1        |
> > > | 0.8         | MARA ($ \tau = 0.80 $)      | 5635±249    | 1407±123   | 0.766±0.008    | 36±3        |
> > > | 0.8         | MARA ($ \tau = 0.85 $)      | 5502±309    | 1426±63    | 0.769±0.006    | 34±3        |
> > > ||||
> > > | 0.85        | REINVENT               | 1098±88     | 4360±257   | 0.721±0.016    | 8±1         |
> > > | 0.85        | MARA ($ \tau = 0.80 $)      | 1235±130    | 3943±303   | 0.733±0.009    | 8±1         |
> > > | 0.85        | MARA ($ \tau = 0.85 $)      | 1438±126    | 4230±401   | 0.725±0.008    | 8±1         |

---

> > > > ### Author Response · Authors · 2025-11-24
> > > >
> > > > > Q14) "In this work, we provide an in-depth understanding of the KL-regularized RL objective, particularly in terms of its diversity" -> could you summarize it in a few words?
> > > >
> > > > Many thanks for this recommendation. We have added to the updated pdf:
> > > > - The regularization divergence and the divergence implicitly minimized between policy and target distribution are distinctive things, and we cannot assume regularization divergences’ properties to translate directly to the one implicitly minimized
> > > > - The regularization divergence, regularization strength and reward function all play a role in the shape of the target distribution. Forward and reverse KL induce different targets.
> > > > - Irrespective of the implicitly minimized divergence ($D(\pi, G)$), we will not have a reward mode covering policy if the target distribution itself does not cover the modes.
> > > > - Specifically, for the reverse-KL regularized case, two common training settings that result in unimodal target distributions are (i) differences in rewards with approximately equal $\pi_{ref}$, which lead to exponential differences in relative probabilities in $G$, and (ii) same rewards with different $\pi_{ref}$, which result in relatively unlikely (but high-rewarding) samples under $\pi_{ref}$ to remain unlikely even at optimum.
> > > >
> > > > -----
> > > >
> > > > ## General writing errors / typos
> > > >
> > > > We apologize for these errors. We have gone through and fixed all raised issues and will include the fix in the updated paper.
> > > >
> > > > -----
> > > >
> > > > [1] He, Andre Wang, Daniel Fried, and Sean Welleck. "Rewarding the unlikely: Lifting grpo beyond distribution sharpening." Proceedings of the 2025 Conference on Empirical Methods in Natural Language Processing. 2025.
> > > >
> > > > [2] Tang, Yunhao, et al. "Optimizing language models for inference time objectives using reinforcement learning." arXiv preprint arXiv:2503.19595 (2025).
> > > >
> > > > [3] Dang, Xingyu, et al. "Weight ensembling improves reasoning in language models." arXiv preprint arXiv:2504.10478 (2025).
> > > >
> > > > [4] Naesseth, Christian, Fredrik Lindsten, and David Blei. "Markovian score climbing: Variational inference with KL (p|| q)." Advances in Neural Information Processing Systems 33 (2020): 15499-15510.
> > > >
> > > > [5] Khalifa, Muhammad, Hady Elsahar, and Marc Dymetman. "A distributional approach to controlled text generation." arXiv preprint arXiv:2012.11635 (2020).
> > > >
> > > > [6] Jhaveri, Yash, et al. "Convergence Theorems for Entropy-Regularized and Distributional Reinforcement Learning." arXiv preprint arXiv:2510.08526 (2025).
> > > >
> > > > [7] Olivecrona, Marcus, et al. "Molecular de-novo design through deep reinforcement learning." Journal of cheminformatics 9.1 (2017): 48.
> > > >
> > > > [8] Guo, Jeff, and Philippe Schwaller. "Beam Enumeration: Probabilistic Explainability For Sample Efficient Self-conditioned Molecular Design." The Twelfth International Conference on Learning Representations.

---

> > > > > ### Comment · Reviewer_K4Yn · 2025-11-25
> > > > >
> > > > > Hi,
> > > > >
> > > > > Is it possible that the posted submission is not the most updated one? I was going through it again, but then I noticed that commented changes in the rebuttal such as "RL is the primary way to train models in settings where the correct solution is not known a priori", and the mentioned figures do not show up anywhere.

---

> > > > > > ### Author Response · Authors · 2025-11-26
> > > > > > **Updated submission pdf**
> > > > > >
> > > > > > Dear reviewer,
> > > > > >
> > > > > > We apologize for the confusion. **We have now uploaded the updated PDF** incorporating all of the mentioned changes.
> > > > > >
> > > > > > We deeply appreciate your time and effort in helping us improve this paper, and we look forward to addressing any further comments you may have!
> > > > > >
> > > > > > Many thanks.

---

> > > > > > > ### Comment · Reviewer_K4Yn · 2025-11-26
> > > > > > >
> > > > > > > Thank you for the updated submission. I haven't gone through all the details, yet. In the meantime, I would like to give some initial impressions because I still stand by my original evaluation that the paper is making some confused and unsupported claims, which given its didactic aims can be very problematic. Let me illustrate this by just commenting the abstract.
> > > > > > >
> > > > > > > > It is commonly believed that optimizing the reverse KL divergence results in “mode seeking”, while optimizing forward KL results in “mass covering”, with the latter being preferred if the goal is to sample from multiple diverse modes. We show— mathematically and empirically—that this intuition does not necessarily transfer well to reinforcement learning with reverse/forward KL regularization
> > > > > > >
> > > > > > > The juxtaposition of these sentences makes it look like there is a contradiction between them, whereas in fact these are two completely independent facts.
> > > > > > >
> > > > > > > > Here, the choice of KL regularizer decides the family of optimal target distributions, and optimizing the full objective corresponds to minimizing a divergence between the policy and this target distribution.
> > > > > > >
> > > > > > > At least in the case of RKL, the objective distribution and the divergence being optimized are _both_ a consequence of the KL-regularized reward maximization objective.
> > > > > > >
> > > > > > > > while mode coverage depends primarily on other factors, such as regularization strength
> > > > > > >
> > > > > > > I don’t understand this sentence. The authors need to make explicit what they call a “mode”, which they are using already from the abstract. Are they high reward samples? High probability samples from the reference? High probability samples of the target distribution? From my understanding of the rest of the paper, I think the authors refer to the first, but it is important that the authors make the terminology clear, and ideally more rigorous. Also, what does coverage mean here? Policy mass on samples meeting any of these alternative definitions of modes? Again, from my understanding of the rest of the paper I don't think this is it, but rather some characteristic of the target distribution that I cannot precisely identify. I would ask the authors to also be more precise on this respect.
> > > > > > >
> > > > > > > > Further, we show commonly used settings such as low regularization strength and equal verifiable rewards tend to specify uni-modal target distributions, meaning the optimization objective is by construction non-diverse.
> > > > > > >
> > > > > > > The example in the Figure 4 shows some artificial example in which there is one sample according to the reference that concentrates all of the probability mass, but have you looked empirically at whether this is the case at all with sampling from LLMs? You can for example test whether the collapse of diversity is due to probability differences in the base model by using rejection sampling: generate $n$ samples from the base model and reject all that are below your target $\tau$. Are you left out with a single sample or are there a diverse set of samples? If the latter, do they have overall the same probability or do they span a wide range?
> > > > > > >
> > > > > > > > We leverage these insights to construct a simple yet principled algorithm, which makes minimal changes to reward magnitudes, and theoretically prove that it optimizes for a target distribution which puts high probability over all high-quality sampling modes
> > > > > > >
> > > > > > > Shaping the reward distribution to be uniformly large at high reward regions (which I would prefer to "high-quality sampling modes") is an interesting idea! The fact that this could be a desirable target does not need to predicate on the alleged "unimodality" of target distribution. The narrative of the paper could be largely independent of the remarks in the first part, especially those that are disconnected from the rest of the paper (e.g. remarks 3.2, 3.3, and 3.4). However, if the paper is focused -as it currently is starting from the title- that the collapse of diversity in KL regularized RL is due to the shape of the target distribution, then this claim should be better supported.

---

> > > > > > > > ### Comment · Reviewer_K4Yn · 2025-11-27
> > > > > > > >
> > > > > > > > >> Further, we show commonly used settings such as low regularization strength and equal verifiable rewards tend to specify uni-modal target distributions, meaning the optimization objective is by construction non-diverse.
> > > > > > > >
> > > > > > > > > The example in the Figure 4 shows some artificial example in which there is one sample according to the reference that concentrates all of the probability mass, but have you looked empirically at whether this is the case at all with sampling from LLMs? You can for example test whether the collapse of diversity is due to probability differences in the base model by using rejection sampling: generate samples from the base model and reject all that are below your target . Are you left out with a single sample or are there a diverse set of samples? If the latter, do they have overall the same probability or do they span a wide range?
> > > > > > > >
> > > > > > > > Another thought on this: [1] already did this some form of this rejection-sampling analysis by measuring whether RL-trained models and the base models can generate at least one correct response out of $k$ samples (pass@k) . If the lack of diversity in KL-regularized RL training model was coming from the shape of the base distribution, then we should see that both the base and the RL model should equally benefit from additional samples, but that is not the case: RL models generate much less diverse samples than the base distribution which is why the latter "catches up" in performance at higher k.
> > > > > > > >
> > > > > > > > [1] Yue et al. 2025. Does Reinforcement Learning Really Incentivize Reasoning Capacity in LLMs Beyond the Base Model? https://arxiv.org/abs/2504.13837

---

> > > > > > > > > ### Author Response · Authors · 2025-12-03
> > > > > > > > >
> > > > > > > > > We thank the reviewer for their comments. **We believe the reviewer’s major concerns are with the information conveyed in the abstract**. **We have revised it majorly, which now should address all concerns.** We have also provided additional experiments as asked.
> > > > > > > > >
> > > > > > > > > Taken together, these changes have substantially improved clarity in service of this paper serving as a didactic discussion about the role of regularization in the regularized RL objective.
> > > > > > > > >
> > > > > > > > > We address individual comments below.
> > > > > > > > >
> > > > > > > > > > Juxtaposition of “optimizing the reverse / forward KL divergence resulting in mode seeking / mass covering” and “reinforcement learning with reverse/forward KL regularization” makes it look like there is a contradiction, whereas in fact these are two completely independent facts.
> > > > > > > > >
> > > > > > > > > Thank you for the suggestion. To clarify, we are making a juxtaposition to point out a point of possible confusion in the community. From our experiences speaking with practitioners, and in a number of existing works [1, 2, 3], the regularizer choice directly motivates diversity through the KLs’ mode-seeking / mass-covering intuitions, without a deep discussion of, for instance, how the reward and/or reference modes affect the optimization target or gradient properties.
> > > > > > > > >
> > > > > > > > > For instance, [1] states the following motivation: _“Typically, reverse KL divergence is a default choice. However, the mode-seeking behavior will lead to low diversity in the generations. Therefore, to balance the alignment performance and diversity, we consider a more broad class of divergence regularization”_. [2] states: _“the community has almost universally adopted the standard reverse-KL divergence, whose well-established mode-seeking nature theoretically forces the policy to converge on a single high probability solution.”_
> > > > > > > > >
> > > > > > > > >
> > > > > > > > > Our goal is precisely to provide more nuance here, specifically:
> > > > > > > > > - We would like the policy to put high mass over all high reward regions
> > > > > > > > > - As currently defined, target distributions do not do this, thus better optimization fundamentally cannot help achieve this
> > > > > > > > > - We can construct better targets by changing the reward and/or changing the regularizer to the ref policy. However, naively changing the regularizer from reverse to forward KL [1,2,3] does not fundamentally address the above issue.
> > > > > > > > > - We instead propose to change the reward so the target puts substantial mass over all high reward regions
> > > > > > > > >
> > > > > > > > > We re-wrote our abstract and introduction in a major way to convey this more clearly.
> > > > > > > > >
> > > > > > > > >
> > > > > > > > > > Re: “The choice of KL regularizer decides the family of target distributions, and optimizing the full objective corresponds to minimizing a divergence”. At least in the case of RKL, the objective distribution and the divergence being optimized are both a consequence of the KL-regularized reward maximization objective.
> > > > > > > > >
> > > > > > > > > Thank you for the comment. We have fixed the phrasing to make it less confusing in the updated abstract.
> > > > > > > > >
> > > > > > > > > > Re: “mode coverage depends primarily on other factors, such as regularization strength”. The authors need to make explicit what they call a “mode” (e.g. in “while mode coverage depends primarily on other factors, such as regularization strength”). I would ask the authors to also be more precise on this respect.
> > > > > > > > >
> > > > > > > > > Indeed, we refer to high reward samples as the “mode”, which we already define in Definition 3.5. We thank the reviewer for pushing this to be rigorous. To this end:
> > > > > > > > > - We have updated definition 3.5 (defining “reward multimodal”) in the paper to be more formal
> > > > > > > > > - We refrained from using “unimodal” in the abstract, instead referring to it as “mass concentrating on a single high-reward region”
> > > > > > > > >
> > > > > > > > > This should clarify any additional confusion about “modes”.
> > > > > > > > >
> > > > > > > > >
> > > > > > > > > -----
> > > > > > > > > [1] Wang, Chaoqi, Yibo Jiang, Chenghao Yang, Han Liu, and Yuxin Chen. 2024. BEYOND REVERSE KL: GENERALIZING DIRECT PREFERENCE OPTIMIZATION WITH DIVERSE DIVER- GENCE CONSTRAINTS.
> > > > > > > > >
> > > > > > > > > [2] Omura, Motoki, Yasuhiro Fujita, and Toshiki Kataoka. 2025. “Entropy Controllable Direct Preference Optimization.” arXiv:2411.07595. Version 2. Preprint, arXiv, June 13. https://doi.org/10.48550/arXiv.2411.07595.
> > > > > > > > >
> > > > > > > > > [3] Li, Long, Jiaran Hao, Jason Klein Liu, et al. 2025. “The Choice of Divergence: A Neglected Key to Mitigating Diversity Collapse in Reinforcement Learning with Verifiable Reward.” arXiv:2509.07430. Version 2. Preprint, arXiv, October 17. https://doi.org/10.48550/arXiv.2509.07430.

---

> > > > > > > > > > ### Author Response · Authors · 2025-12-03
> > > > > > > > > >
> > > > > > > > > > > Re: “We show commonly used settings such as low regularization strength and equal verifiable rewards tend to specify uni-modal target distributions”, have you looked empirically at whether this is the case at all with sampling from LLMs? You can for example test [this] by using rejection sampling
> > > > > > > > > >
> > > > > > > > > > Thank you for the suggestion. **We ran additional experiments and added results to Appendix C.4**. In summary:
> > > > > > > > > > - In the verifiable “1-2” task setting, the correct samples overwhelming produce “1” instead of “2”, indicating a skewed base distribution.
> > > > > > > > > > - In the Q&A setting, we sample 1k+ responses from the base model and filter for high reward answers. We see the difference in their base model log probability can be up to 7.76, corresponding to a probability ratio of ~2345x more likely.
> > > > > > > > > > - If we account for both the difference in reference probability and the reward function (with $\beta = 0.01$), the most likely response is over 10^40 times more likely than the 16th most likely response, meaning that the resulting distribution is not multimodal.
> > > > > > > > > >
> > > > > > > > > > Taken together, there is strong empirical evidence suggesting that when sampling from LLMs, the shape of the reference model and reward function induce highly skewed target distributions.
> > > > > > > > > >
> > > > > > > > > >
> > > > > > > > > > > Another thought on [unimodal target distribution]. [1] measures pass@k for base models vs. RL-trained models. If the lack of diversity in KL-regularized RL training model was coming from the shape of the base distribution, then we should see that both the base and the RL model should equally benefit from additional samples, but that is not the case: RL models generate much less diverse samples than the base distribution which is why the latter "catches up" in performance at higher k.
> > > > > > > > > >
> > > > > > > > > > Rephrasing to ensure we understand the question correctly: assuming binary rewards, the optimal policy should have the same relative probability between all correct answers as the reference distribution. However, Figure 1 and 2 in [4] suggests the RL-trained policy has _less_ coverage over answers than the base model. This is incongruous with the prediction in Remark 4.3 that for samples with the same reward, their probability ratio is simply the reference probability ratio, $\pi_{ref}(y_1) / \pi_{ref} (y_2)$.
> > > > > > > > > >
> > > > > > > > > > This is an interesting question! However, we do not think this is incongruous:
> > > > > > > > > > - To start, pass@k is not a good metric for the type of diversity / multimodality we discuss. For instance: a model with all mass collapsed on a single correct answer will have perfect pass@k, but is not considered diverse per our definition – our work instead deals with whether or not the policy puts mass over multiple equally correct answers.
> > > > > > > > > > - Further, the optimal policy is not necessarily reached in practice. The policy obtained through RL can be **even less diverse** than the theoretically optimal one. Our predictions and method is only about the optimal target distribution. We already discuss this in the conclusion (L525)
> > > > > > > > > > - A hypothesis for how learning dynamic might drive sub-optimality: since we are doing stochastic optimization, if a minibatch contains only a single kind of correct answer (out of multiple kinds), the policy gradient / score function optimization update $R(y) \nabla \log \pi (y)$ will increase its probability, and also makes it less likely subsequently to find other answers.
> > > > > > > > > >
> > > > > > > > > > > Shaping the reward distribution to be uniformly large at high reward regions (which I would prefer to "high-quality sampling modes") is an interesting idea!.. However, if the paper currently is starting from the title- that the collapse of diversity in KL regularized RL is due to the shape of the target distribution, then this claim should be better supported.
> > > > > > > > > >
> > > > > > > > > > We have better supported the diversity collapse claim in the newly added experiments. We have also updated the writing to make the didactic discussions clearer.
> > > > > > > > > >
> > > > > > > > > > -----
> > > > > > > > > >
> > > > > > > > > > [4] Yue et al. 2025. Does Reinforcement Learning Really Incentivize Reasoning Capacity in LLMs Beyond the Base Model? https://arxiv.org/abs/2504.13837

---

### Official Review · Reviewer_v1oM · 2025-10-26

**Soundness:** 2
**Presentation:** 1
**Contribution:** 1
**Rating:** 2
**Confidence:** 4

**Summary:**

This paper studies the effect of KL regularization in the objective function of LLM post-training from the perspective of diversity, e.g., the multi-modals in the learned distributions from the final policy. The authors compare the optimal policy under the reverse and forward KL regularizations under the KL-regularized objective in LLM post-training. Based on the analysis, they proposed to put equal weights on all high-quality samples to encourage the multi-modal target. Experiments are conducted on several LLM post-training settings.

**Strengths:**

1. Understanding the learning behaviors, e.g., the multi-modality of the learned distribution, of the KL-regularized objective function in LLM post-training is an important research problem.

2. The usage of illustration improves the readability, although in simple settings.

**Weaknesses:**

1. **The scope of the paper is questionable**. The title of this paper targets the general RL, but I am surprised that the scope of this paper only focuses on the KL-regularized objective function used in LLM post-training. There are a flurry of prior RL works on pure KL-regularized RL, such as MaxEnt RL, Soft Actor Critic, and natural policy gradient, that focus on the MDP setting of the general RL setting. By contrast, the KL-regularized objective in LLM only assumes a specific contextual bandit setting with iid samples, which is far less representative of the general RL setting. Using such a title is very misleading. Researchers in LLM post-training should respect prior works in the general KL-regularized policy gradient methods if they really want to target a general research problem. Alternatively, it is required to narrow down the scope and title of the paper to the LLM setting without causing confusion, which is just a specific application of general RL in the language scenario. Lots of analyses are very divergent.

2.  **Less convincing theoretical analysis**.
* The paper relies on the fact that reverse and forward KL have the effect of mass covering or mode seeking, but I do not think they are established in a fundamental way. The authors are suggested to provide sufficient references, either from information theory or statistics; otherwise, the subsequent analysis would be less convincing. Also, some illustrations on experiments are too simple to make a rigorous conclusion. This issue occurs many times across the paper.
* Is the diversity the sole objective in the policy optimization? In statistics, the reason why people use MLE or equivalent KL divergence is that it ensures statistical efficiency, asymptotic unbiasedness, and consistency, often on iid samples. However, if you manually introduce the sample selection bias by focusing on the diversity to encourage multi-modality, there is definitely some price to pay. However, from the theoretical perspective, I did not find such a deep analysis or conclusion regarding this point. After reading Line 102, does it mean that in LLM the foundation models are often less expressive such that we need to analyze the difference of two KL divergences? A lot of details are unclear to me.
* This paper makes a lot of strong claims, such as ‘diversity collapse is a natural consequence of solving the RL problem.’ It is very risky to accept these claims without finding rigorous proof or strong empirical evidence or articulating the specific setting of the RL problem they are studying among the whole RL literature.
* In Eq. 5, there is no closed-form solution under forward KL divergence, where $\Lambda$ is the negative of the Lagrangian multiplier. However, it does not necessarily mean that the optimizers under the two KL divergences are completely different in the general case (line 163)!
* The analysis in Section 4 is performed under very specific scenarios. Although sometimes they provide some insights, in general they are very unlikely to hold in the training process. Such an analysis is fragile and less convincing. The result in Proposition 4.1 is trivial. The logic in Section 4 is very hard to follow.

3. **Unclear and informal writing**. I personally think a lot of writing is not formal or rigorous.
* What is the definition of ‘solution distribution’? I understand what it is by guess, but the authors did not give a definition or explanation when they used it many times.
* What is the approximate inference in Line 53? I do not think it is a scientific word.
* In Line 92. What is the definition of two-parameter Gaussian? Is the foundation model as an example of a flexible distribution in line 93?
* It is inaccurate to say we aim to maximize a reward function in Line 107. Instead, we are maximizing the average rewards or the expectation of the returns. What is the definition of "off-support samples in Line 264. There are lots of expressions that are not formal, which is discouraged to me.
* There are also many grammar errors, such as ‘This let’ in Line 232 and ‘KL exhibit’ in Line 91. The writing needs substantial improvements.

4. **Unclear empirical validations**. Since the paper focuses on the improvement of diversity, I could not find a detailed explanation about the metrics of diversity. Learning curves in Figure 7 are blurred, and thus it is hard to make a clear conclusion on it. Also, it makes more sense to me if the LLM focuses on the diversity; the efficiency would be reduced to some extent. Sometimes this price is worthy to pay, but I could not find such an analysis in the experiments.

**Questions:**

Please see the weaknesses.

---

> ### Author Response · Authors · 2025-11-24
>
> We thank the reviewer for their illuminating comments and time. We are happy the reviewer thinks this paper provides better understanding of learning behaviour for KL-regularized objectives, that this is a fundamentally important research problem, and that illustration provides readability.
>
> We address all questions below.
>
> ## Weaknesses
>
> ### 1. Scope of the paper is questionable.
>
> > The title of this paper targets the general RL, but I am surprised that the scope of this paper only focuses on the KL-regularized objective function used in LLM post-training. There are a flurry of prior RL works on pure KL-regularized RL, such as MaxEnt RL, Soft Actor Critic, and natural policy gradient, that focus on the MDP setting of the general RL setting. By contrast, the KL-regularized objective in LLM only assumes a specific contextual bandit setting with iid samples, which is far less representative of the general RL setting. Using such a title is very misleading. Researchers in LLM post-training should respect prior works in the general KL-regularized policy gradient methods if they really want to target a general research problem.
>
> We thank the reviewer for the suggestion to connect with previous MDP works. We have included additional discussions of related works including [1], and max entropy RL [2,3,4]. We have also clarified we are in a non-MDP / contextual bandits setting upfront in the introduction to avoid confusion. We are amenable to change the title to e.g. “KL-Regularized Reinforcement Learning for Generative Models is Designed to Mode Collapse” if the reviewer finds it more appropriate.
>
> That said, we believe the paper’s focus on the contextual bandit setting is nevertheless very relevant, as this setting applies broadly to LLMs and modern generative models. Our analysis and algorithmic contribution is novel and valuable for this broad area, and moreover this style of analysis can still provide insights for analyzing the MDP seeing, for instance as done in [5].
>
>
> ### 2. Less convincing theoretical analysis
>
> > The paper relies on the fact that reverse and forward KL have the effect of mass covering or mode seeking, but I do not think they are established in a fundamental way. The authors are suggested to provide sufficient references, either from information theory or statistics.
>
> The mode seeking / mass covering behaviour of reverse / forward KL is a fundamental observation in variational inference, see for instance the Bishop textbook [6], chapter 10.1, or the Murphy textbook [7], chapter 21.2. We already cite both in L102.
>
> > Is the diversity the sole objective in the policy optimization? After reading Line 102, does it mean that in LLM the foundation models are often less expressive such that we need to analyze the difference of two KL divergences?
>
> The point of our paper is to view RL as used in generative models as  a variational inference problem: i.e. the policy distribution is attempting to match some target distribution. Due to the mode seeking / mass covering behaviour of the two KLs, there is a common belief that changing the KL regularizer can promote different behaviours which motivates various previous works.
>
> The main point in our paper is that regardless of the divergence being optimized / used as regularization, if the _target distribution_ is unimodal, we will not get a multimodal distribution from optimizing the objective. We point out that this is in fact the case in practice with commonly used hyperparameter settings and reward functions, which is an issue and should be considered as part of the on-going discussion about diversity collapse in post-trained generative models.

---

> > ### Author Response · Authors · 2025-11-24
> >
> > > This paper makes a lot of strong claims, such as ‘diversity collapse is a natural consequence of solving the RL problem.’ It is very risky to accept these claims without finding rigorous proof or strong empirical evidence or articulating the specific setting of the RL problem they are studying among the whole RL literature.
> >
> > To be precise, the claim is that the _regularized_, reward maximizing objective has unimodal optimal policies _with respect to the reward function modes_, for **commonly used hyperparameters** (as stated in L70).
> >
> > Entropy / KL regularizers try to improve policy diversity by having “soft”, stochastic optimal policies. However, we prove in section 4 (remark 4.2 and 4.3) that with commonly used hyperparameters, the optimal policy still puts probability mass disproportionately on actions that are either (i) only slightly better (see [updated figure](https://ibb.co/dw4jZSmq)), or (ii) have higher likelihood under the reference reference (see [updated figure](https://ibb.co/DHTVsJtz)). Diversity collapse is a natural corollary of this observation (since we are not distributing probability mass amongst all good actions).
> >
> > We have softened L72 writing to “We show that the optimal policy is often by definition unimodal… meaning diversity collapse is a natural consequence of correctly solving the regularized RL problem as defined by commonly used hyperparameters”. We have also included the figures above in-text for greater clarity. Let us know if you would prefer further edits.
> >
> > N.B. in the unregularized case where the goal is simply to find the reward maximizing policy, there can exist multiple optimal policies, and special care needs to be taken to ensure that the final policy is _not_ a deterministic policy concentrated at a single high-reward mode [10] (this also relates to fundamental results in MDP theory, such as solution for stationary deterministic policies is same as for stationary randomized policies, proposition 6.2.1 of [8]).
> >
> >
> > > In Eq. 5, there is no closed-form solution under forward KL divergence… However, it does not necessarily mean that the optimizers under the two KL divergences are completely different in the general case (line 163)!
> >
> > The two optimal distributions _are_ different in the general case, as their density functions (Eq 2 and 5) are different. For them to have the same optimal distribution, the two equations have to be the same, which is not true in the general case. If we mis-understood this comment we are open to discussing this further!
> >
> > Also note Equation 5 does have a solution, but $\Lambda$ is not known a priori and needs to be solved. $\Lambda$ plays the role of normalizing the distribution (Eq 5), just as $\zeta$ does for the reverse KL case (Eq 2). We have updated our writing to clarify this.
> >
> >
> > > The analysis in Section 4 is performed under very specific scenarios. In general they are very unlikely to hold in the training process. Such an analysis is fragile and less convincing. The result in Proposition 4.1 is trivial. The logic in Section 4 is very hard to follow.
> >
> > We respectfully disagree that section 4’s analysis is trivial and unlikely to hold in training. Proposition 4.1 is a generic way to analyze any (reverse-KL regularized) target distribution. We then apply this to very commonly used settings which many LM training today use. Further, our analysis of the target distribution holds true for _any_ training processes as it is **provably the global optimizer** for the objective being optimized. That is, for **any** training run, as long as the objective is correctly optimized, the policy will move toward the target distribution stated in our paper. Further, even if this result may be obvious in hindsight, it is often overlooked in the community, in which case our work has value in helping to provide some clarity to the currently on-going discussion.

---

> ### Author Response · Authors · 2025-11-24
>
> ### 3. Unclear and informal writing
>
> > What is the definition of ‘solution distribution’? I understand what it is by guess, but the authors did not give a definition or explanation when they used it many times.
>
> It is defined in L113. It refers to the target i.e. optimal distribution of (regularized) policy optimization.
>
> > What is the approximate inference in Line 53? I do not think it is a scientific word.
>
> Approximate inference refers to methods for estimating a probability distribution when computing the exact posterior is intractable (see Bishop Chapter 10 [6], Murphy Chapter 21 [7] , Goodfellow Chapter 19 [9]. It also has a wikipedia page https://en.wikipedia.org/wiki/Approximate_inference ). Variational inference is one way of doing approximate inference, which we leverage in our paper.
>
> We updated L53 to say “we analyze KL-regularized reward maximization through tools from variational inference (VI) to find and dissect optimal policies for different choices of KL regularization” instead. We hope this provides greater clarity.
>
> > In Line 92. What is the definition of two-parameter Gaussian?
>
> A 1D gaussian distribution has two parameters: mean and variance.
>
> > Is the foundation model as an example of a flexible distribution in line 93?
>
> Yes.
>
> > It is inaccurate to say we aim to maximize a reward function in Line 107. Instead, we are maximizing the average rewards or the expectation of the returns.
>
> Many thanks for pointing this out, we have fixed this in the updated pdf.
>
> > What is the definition of "off-support samples in Line 264.
>
> We refer to samples $y$ which have small $\pi_{ref}(y)$. We have changed all instances of “off support” to “low support” to be precise (as off support usually means $\pi_{ref}(y) = 0$). Thank you for pointing this out.
>
>
> ### 4. Unclear empirical validations.
>
> >  Since the paper focuses on the improvement of diversity, I could not find a detailed explanation about the metrics of diversity.
>
> We provided an explanation for the diversity metrics in Appendix C.3 and C.4, and referenced them in main text L419 and L444. We have added an additional short sentence section 6.2 and 6.3 to clarify further, and we are happy to expand if anything else is unclear.
>
> > Learning curves in Figure 7 are blurred, and thus it is hard to make a clear conclusion on it. Also, it makes more sense to me if the LLM focuses on the diversity; the efficiency would be reduced to some extent. Sometimes this price is worth paying, but I could not find such an analysis in the experiments.
>
> Figure 7 says with regular regularized reward maximization (grey), policy tends to collapse towards one out of the two solutions. With MARA (blue), policy preserves better coverage over both equally correct solutions. This is seen in higher *valid* answer diversity (Fig 7a, middle, blue being higher).
>
> In the same task, reward augmentation does slow down the increase in average reward slightly, which is shown in Fig 7a (blue being slightly below grey), right, but it still reaches a good solution eventually.
>
>
> Lastly, re: empirical validations, we have added additional baselines and ablations in the Global Response for the reviewer’s consideration! Please let us know if anything else is unclear.
>
> -----
>
>
> [1] Chan, Alan, et al. "Greedification operators for policy optimization: Investigating forward and reverse kl divergences." Journal of Machine Learning Research 23.253 (2022): 1-79.
>
> [2] Ziebart, Brian D., et al. "Maximum entropy inverse reinforcement learning." Aaai. Vol. 8. 2008.
>
> [3] Haarnoja, Tuomas, et al. "Reinforcement learning with deep energy-based policies." International conference on machine learning. PMLR, 2017.
>
> [4] Huang, Shiyu, et al. "Svqn: Sequential variational soft q-learning networks." International Conference on Learning Representations. 2019.
>
> [5] Grill, Jean-Bastien, et al. "Monte-Carlo tree search as regularized policy optimization." International Conference on Machine Learning. PMLR, 2020.
>
> [6] Bishop, Christopher M., and Nasser M. Nasrabadi. Pattern recognition and machine learning. Vol. 4. No. 4. New York: springer, 2006.
>
> [7] Murphy, Kevin P. Probabilistic machine learning: an introduction. MIT press, 2022.
>
> [8] Puterman, Martin L. "Markov Decision Processes: Discrete Stochastic Dynamic Programming." (1994).
>
> [9] Goodfellow, Ian, et al. Deep learning. Vol. 1. No. 2. Cambridge: MIT press, 2016.
>
> [10] Jhaveri, Yash, et al. "Convergence Theorems for Entropy-Regularized and Distributional Reinforcement Learning." arXiv preprint arXiv:2510.08526 (2025).

---

### Official Review · Reviewer_mZ9a · 2025-10-31

**Soundness:** 4
**Presentation:** 4
**Contribution:** 3
**Rating:** 8
**Confidence:** 4

**Summary:**

This paper studies KL regularization in RL finetuning of language models from first principles. This paper lays out various facts about the global solutions of the forward and reverse kl regularization + RL finetuning process:

1) RL + reverse KL regularization is equivalent to minimizing reverse kl with its globally optimal solution (known previously)
2) RL + forward Kl regularization does not optimize any form of forward KL.
3) In RL + forward KL, the beta coefficient acts as an exponential decay / growth term in the solution. Standard values can essentially make the best solution orders of magnitude more likely than other solutions. This is known but often overlooked.
4) When rewards are equal, off policy solutions are discouraged. This is also obvious (looking at equation 2), but still overlooked.

Finally, for KL regularization + RL finetuning, they provide a simple heuristic to boost multi-modality and present experimental results verifying their approach.

**Strengths:**

I think the summary^ is a fair assessment of the paper's strengths. It is presents simple results in a very clear and easy to read manner. According to me, the heuristic presented seems to be not that significant in terms of usability in frontier models for example. But I could be wrong on that. For the theoretical results, they are simple in hindsight, but are often overlooked in the literature. Remark 3.3 seems to be the most non trivial and novel result in my opinion.

Overall I enjoyed reading this paper and I think this paper should be accepted.

**Weaknesses:**

The only weakness I see is a lack of strong baselines. LLM post training is not my primary area of research but surely their have to be many multi-modality inducing algorithms for post training (RL exploration, gflownets, feature  covariance maximization, etc). I do not expect your algorithm to be better than them, but a comparison might still be useful for readers.

**Questions:**

See weaknesses

---

> ### Author Response · Authors · 2025-11-24
>
> We thank the reviewer for the insightful comments! We are happy the reviewer finds the results simple and easy to read, that our conceptual contribution, even if simple in hindsight, is often overlooked in practice, and that our target solution for the forward KL regularized case (Remark 3.3) is non-trivial and novel.
>
> We address the comments below.
>
> ## Weaknesses and Questions
>
> > The heuristic presented seems to be not that significant in terms of usability in frontier models
>
> We gently push back here. RL post-training is an essential component for frontier models, and diversity collapse remains an issue. Our paper provides fundamental facts which should be meaningful for any regularized RL training procedures, and MARA provides one example of **principally** optimizing for a by-construction multimodal target distribution.
>
> To be comprehensive, we’ve added more discussions about potential limitations of MARA in the conclusion:
>
> - MARA addresses issues with the _shape_ of the target distribution, but we do not make theoretical claims about optimization efficiency toward the target
> - The choice of $\tau$ may be less obvious in continuous, unbounded reward settings (though we do have an approximation as used in Sec 6.2, and further ablated this in the Global Response)
>
> We think these will be useful directions for future works to build upon. Overall, we think MARA is a step towards designing fundamentally better objectives for long-term RL training regardless of the model size.
>
>
> > Lack of strong baselines
>
> We have run additional baselines and included them below and updated the pdf to include them. Please let us know if there are other baselines you had in mind.
>
> ### Creative Q&A (section 6.2, Table 1)
>
> We added additional LLM diversity-inducing baselines for the results in section 6.2, including entropy regularization, rewarding the unlikely [1], Best-of-N training [2], and weight ensembling [3]. We found that MARA remains the most performant in 3/4 metrics.
>
> Please let us know if you think any other baselines would be appropriate!
>
>
> | Model              | In-dist Reward (↑) | Out-dist Reward (↑)           | Ngrams EAD (↑)              | SemDiv (↑)            | Mean Distinct  (↑)         |
> |--------------------|---------|---------------------|------------------|--------------------|--------------------|
> | Base Model         | 10.94   | 1.166 ± 0.076       | 0.413 ± 0.015     | 0.220 ± 0.009      | 4.01 ± 0.254       |
> | GRPO               | 14.8    | 1.317 ± 0.102       | 0.497 ± 0.014     | 0.193 ± 0.009      | 3.96 ± 0.249       |
> | RLOO               | 15.56   | 1.280 ± 0.100       | 0.514 ± 0.014     | 0.192 ± 0.008      | 3.88 ± 0.243       |
> | Entropy            | 1.44    | 0.786 ± 0.073       | 0.267 ± 0.009     | **0.228 ± 0.009**  | 3.45 ± 0.193       |
> | Unlikely       | 10.04   | 1.381 ± 0.114       | 0.532 ± 0.015     | 0.191 ± 0.008      | 4.24 ± 0.239       |
> | BoN training    | **16.88** | 0.596 ± 0.055     | 0.541 ± 0.010     | 0.162 ± 0.008      | 2.29 ± 0.173       |
> | Ensembling          | -       | 1.143 ± 0.086       | 0.438 ± 0.014     | 0.211 ± 0.010      | 4.19 ± 0.269       |
> | MARA (rev)         | 15.42   | 1.451 ± 0.103       | 0.543 ± 0.014     | 0.186 ± 0.008      | 4.14 ± 0.233       |
> | MARA (fwd)         | 15.33   | **1.604 ± 0.113**   | **0.568 ± 0.012** | 0.193 ± 0.009      | **4.62 ± 0.258**   |
>
>
>
> -----
>
> [1] He, Andre Wang, Daniel Fried, and Sean Welleck. "Rewarding the unlikely: Lifting grpo beyond distribution sharpening." Proceedings of the 2025 Conference on Empirical Methods in Natural Language Processing. 2025.
>
> [2] Tang, Yunhao, et al. "Optimizing language models for inference time objectives using reinforcement learning." arXiv preprint arXiv:2503.19595 (2025).
>
> [3] Dang, Xingyu, et al. "Weight ensembling improves reasoning in language models." arXiv preprint arXiv:2504.10478 (2025).

---

### Official Review · Reviewer_W9ms · 2025-10-31

**Soundness:** 3
**Presentation:** 2
**Contribution:** 3
**Rating:** 6
**Confidence:** 3

**Summary:**

This paper looks at the behavior of policy gradient optimization with respect to the forward vs. reverse KL-regularized expected reward, where it is demonstrated both theoretically and empirically that (i) either choice of divergence can, in the right circumstances, cover the various modes of the target distribution and (ii) there are circumstances in which neither will appropriately cover the various modes. The authors propose a modification to the reward function, MARA, such that policy gradient optimization with respect to the forward or reverse KL-regularized expected modified reward assigns high probability to all actions that induce a sufficiently high reward.

**Strengths:**

This paper does a good job highlighting that common intuitions about the mode-seeking vs. mass-covering nature of different divergence choices does not obviously result in the expected behavior that is often highlighted by 1d examples. Indeed, for expected reward regularized divergence minimization, the authors demonstrate in a convincing way both formally and through experiments how the choice of reference policy, regularization strength, the shape of the reward function are all highly relevant factors in determining whether the learned policy covers the various modes of the reward function. I found the authors solution to this (mode anchored reward augmentation) to be an original one and well motivated by the paper's exposition. The paper's experiments are well constructed and show on a variety of pertinent RL problems that MARA generally improves diversity of the resulting policy.

**Weaknesses:**

The MARA reward function itself feels a bit unsatisfying in that it requires knowledge about reasonable choices of tau (and therefore some knowledge/belief about the distribution of attainable reward values). I wonder to what extent this makes MARA suitable in settings where rewards can be potentially unbounded in magnitude (e.g., certain Atari games). The need to have a high quality anchor example also feels like a limitation on the problems that MARA may be applicable to, e.g., problems where the reward signal is very sparse, or where high reward examples are rare a priori. Further, it seems that redefining the reward to a constant for all sufficiently high reward examples may not be appropriate for all problems.

**Questions:**

Do you have an explanation for why MARA has lower "semantic embedding" diversity than the baseline?

How sensitive is the behavior of the MARA learned policy to choice of tau? It would be helpful to have a figure or result dedicated to this (I was not able to find it in the paper, please point it out to me if I overlooked it). Also, how did you go about selecting the value of tau in experiments?

In the REINVENT task, is there a good explanation for why using MARA as a drop-in replacement did not lead to improvement on the two diversity related metrics? Have you compared the number of, e.g., ECFP4-based Tanimoto clusters for compounds sampled from REINVENT vs. REINVENT with MARA objective?

How sensitive is MARA to the choice of the anchor z?

---

> ### Author Response · Authors · 2025-11-24
>
> ## Weaknesses
>
> > W1: MARA reward function itself feels a bit unsatisfying in that it requires knowledge about reasonable choices of tau. I wonder to what extent this makes MARA suitable in settings where rewards can be potentially unbounded in magnitude
>
> We’d like to point out there are many cases where the choice of $\tau$ is obvious:
>
> 1. For binary rewards (Sec. 6.1), the threshold is simply 1 (i.e. all positive answers are above threshold). This has the effect of encouraging all correct answers, rather than collapsing onto a single one.
> 2. Some continuous reward settings a priori known threshold. For instance, in drug discovery (sec 6.3), it is common to want to “generate unique molecules above a certain reward threshold” (e.g. 0.8), to screen molecules for next stage evaluations. MARA is well-suited for this problem (puts mass above some reward threshold with no preference between minor changes in rewards above it).
>
> The reviewer is correct to point out $\tau$ is less obvious in continuous, unbounded reward settings. Section 6.2 deals with this setting: it is an alignment setting with a human preference reward model. In this case, **we set $\tau$ for each batch as a percentile of the batch’s reward distribution** (we use 90th percentile, and added new ablation results below). We found **this approximation still had an effect to improve diversity**.
>
> We have included this discussion in the conclusion to clearly note the limitation and as a meaningful direction for future works.
>
> > W2: The need to have a high quality anchor example feels like a limitation on the problems that MARA may be applicable to, e.g., sparse reward problems, or where high reward examples are rare a priori.
>
> When no samples achieve $R(y) \geq \tau$, the MARA mechanism does not kick in, reducing the update to the vanilla policy gradient update. Therefore, we do not expect MARA to perform worse than “vanilla” RL, but naturally encourages diversity _when_ multiple above threshold solutions can be discovered.
>
> Specifically, the gradient update for sample $y_i$ _with_ MARA is:
> $( R(z) - \beta (\log \pi_\theta (y_i) - \log \pi_{ref} (z)) \nabla_\theta \log \pi_\theta (y_i)$ , and _without_ MARA is: $( R( y_i ) - \beta (\log \pi_\theta ( y_i ) - \log \pi_{ref} ( y_i )) \nabla_\theta \log \pi_\theta (y_i)$ . See Appendix B.8 for derivation details.
>
> Fundamentally, MARA addresses issues with the global optimum of the objective. An interesting line of future work is how to reach the optimum faster. We've added this as a future direction in the concluding discussion.
>
> > W3: redefining the reward to a constant for all sufficiently high reward examples may not be appropriate for all problems.
>
> The optimal solution for the MARA objective is a policy distribution that puts uniform probability over all “sufficiently good” examples (see Appendix Remark B.2 for the shape of the MARA target distribution). The reviewer is correct to point out this distribution may not be desirable across all applications. However, in the absence of  additional task-specification, it’s less clear if obviously better choices are available?
>
> We believe having uniform likelihood is a simple, domain agnostic choice. Further, by viewing RL through the lens of variational inference, MARA opens up a line of exciting future work on how to design alternative, task-specific shapes of target distributions. We didn’t think it necessary to cover all potential use cases here, however, we’ve noted this as a consideration in the conclusion.

---

> > ### Author Response · Authors · 2025-11-24
> >
> > ## Questions
> >
> > > Why MARA has lower "semantic embedding" diversity than the baseline?
> >
> > MARA has no explicit mechanism for increasing semantic embedding, it simply tries to maintain probability mass over all discovered _good_ answers. Note most post-trained models have lower semantic diversity, hinting at a potential trade-off between semantic-diversity as measured and quality in this task. MARA has no worse semantic diversity than other high-performing baselines.
> >
> > > How sensitive is the behavior of the MARA learned policy to the choice of tau?
> >
> > Note that in many settings (e.g. binary rewards, see W1 response), the choice of tau is obvious (see response to W1 above).
> >
> > We include an ablation with MARA run at a lower $\tau$ for the section 6.2 experiments. We see a slight decrease in performance but overall performance that is still better / competitive with all baselines.
> >
> >
> > | Model                   | Out Dist Reward (↑)   | EAD (↑)           | SemDiv (↑)      | Distinct (↑)    |
> > |-------------------------|---------------------|--------------------|------------------|------------------|
> > | MARA (rev, 0.9)         | 1.451 ± 0.103       | 0.543 ± 0.014      | 0.186 ± 0.008    | 4.14 ± 0.233     |
> > | MARA (fwd, 0.9)         | 1.604 ± 0.113      | 0.568 ± 0.012  | 0.193 ± 0.009    |  4.62 ± 0.258    |
> > | MARA (rev, 0.75)        | 1.498 ± 0.117       | 0.547 ± 0.013      | 0.183 ± 0.008    | 4.41 ± 0.262     |
> > | MARA (fwd, 0.75)        | 1.325 ± 0.097       | 0.508 ± 0.014      | 0.196 ± 0.009    | 4.07 ± 0.243     |
> >
> >
> >
> > > How sensitive is MARA to the choice of the anchor z?
> >
> > The anchor $z$ needs to be an above-threshold sample with high $\pi_{\text{ref}}(y)$. In practice, we do this in a per-batch way to take the highest $\pi_{\text{ref}}(y)$ sample in the batch. Randomly picking the anchor will not work in our preliminary experiments – this makes sense since the point is to anchor on a reward mode, i.e. a sample with high reward _and_ high $\pi_{\text{ref}}(y)$.
> >
> >
> > > In the REINVENT task, is there a good explanation for why using MARA as a drop-in replacement did not lead to improvement on the two diversity related metrics?
> >
> > There are two types of diversity in the molecule generation task. “Local” diversity, characterized by Yield, simply requires generation of different molecules in chemical strings. “Global” diversity (IntDiv1, #Circles) measures more specific differences in molecular sub-structures (i.e. local groups of atoms). MARA simply encourages all discovered good molecules to be well-represented, but does not actively promote specific types of global diversity, which may be a reason for the lack of “global” diversity improvements.
> >
> > It should be noted that in the literature, one often observes a trade off between “local” vs “global” diversity (i.e. higher Yield of unique molecules comes at the cost of these molecules being more globally similar), e.g. as seen in Appendix A.4 of the Practical Molecular Optimization (PMO) benchmark [1]. With MARA, we are able to _maintain_ global diversity while improving local diversity in the form of unique Yield.
> >
> >
> > > In REINVENT, Have you compared the number of, e.g., ECFP4-based Tanimoto clusters for compounds sampled from REINVENT vs. REINVENT with MARA objective?
> >
> > Many thanks for the suggestion. We believe the #Circles metric already well-captures diversity as measured by ECFP4 Tanimoto similarity. In fact, #Circles use fingerprint Tanimoto similarity, then computes a notion of space coverage, which has been shown to be a good diversity metric [2]. We would expect #Circles to perform similarly to fingerprint-based Tanimoto clusters
> >
> > Precisely: starting from the set of generated molecules above the given reward threshold (i.e. 0.80 or 0.85), the “Morgan fingerprints with radius 2” were computed which are roughly equivalent to ECFP4 fingerprints [following the RDKit documentation, 3]. #Circles then quantifies the largest subset of generated molecules, where each molecule is dissimilar to every other molecule by Tanimoto similarity. We impose that all molecules are < 0.25 similarity to each other. We have added this to the updated pdf appendix for clarity.
> >
> >
> > -----
> > [1] Gao, Wenhao, et al. "Sample efficiency matters: a benchmark for practical molecular optimization." Advances in neural information processing systems 35 (2022): 21342-21357.
> >
> > [2] Xie, Yutong, et al. "How Much Space Has Been Explored? Measuring the Chemical Space Covered by Databases and Machine-Generated Molecules." The Eleventh International Conference on Learning Representations.
> >
> > [3] https://www.rdkit.org/docs/GettingStartedInPython.html#morgan-fingerprints-circular-fingerprints

---

### Author Response · Authors · 2025-11-24

We thank all reviewers for their insightful comments which have made this paper better. We are encouraged by the following:

- There is consensus that our paper provides **timely understanding on a currently highly debated and important topic** regarding the behaviour KL-regularized policy gradient (W9ms, v1oM, K4Yn), to **provide insights that are often overlooked** in practice (mZ9a).
- Our **theoretical contribution is non-trivial and novel** (mZ9a, K4Yn).
- Our **algorithmic contribution is original and well-motivated** by the theoretical insights (W9ms, K4Yn).
- Empirically, the **experiments are well-constructed** (W9ms)
- The paper is **well-written** (mZ9a, K4Yn), and **illustrations help to convey intuitions well** (W9ms, v1oM).

We have **addressed all reviewers’ concerns in individual comments**. We also updated our paper pdf incorporating all stated changes. In the interest of not having two paper versions which may cause confusion, we will wait to upload the updated pdf after consolidating any additional reviewer comments during the discussion period.

To summarize the main changes:

-  Added clarifications in the writing, including new figures (response to K4Yn)
- Added more extensive related work section, connecting it with previous works on max entropy RL (v1oM)
- Added additional diversity-inducing baselines for the results in section 6.2, including entropy regularization, rewarding the unlikely [1], Best-of-N training [2], and weight ensembling [3]. We found that MARA remains the most performant in 3/4 metrics.
- Added ablation of the effect of different thresholds of $\tau$ in section 6.2. We find MARA remains competitive.
- Added additional runs of MARA with higher threshold in the drug discovery setting (sec 6.3). We found this significantly increased yield even further when evaluating at a stricter screening threshold

We include all results below.

-----

[1] He, Andre Wang, Daniel Fried, and Sean Welleck. "Rewarding the unlikely: Lifting grpo beyond distribution sharpening." Proceedings of the 2025 Conference on Empirical Methods in Natural Language Processing. 2025.

[2] Tang, Yunhao, et al. "Optimizing language models for inference time objectives using reinforcement learning." arXiv preprint arXiv:2503.19595 (2025).

[3] Dang, Xingyu, et al. "Weight ensembling improves reasoning in language models." arXiv preprint arXiv:2504.10478 (2025).

---

> ### Author Response · Authors · 2025-11-24
>
> # Additional Results
>
> ## Creative Q&A (section 6.2, Table 1)
>
> Added additional diversity-inducing baselines
>
> | Model              | In-dist Reward (↑) | Out-dist Reward (↑)           | Ngrams EAD (↑)              | SemDiv (↑)            | Mean Distinct  (↑)         |
> |--------------------|---------|---------------------|------------------|--------------------|--------------------|
> | Base Model         | 10.94   | 1.166 ± 0.076       | 0.413 ± 0.015     | 0.220 ± 0.009      | 4.01 ± 0.254       |
> | GRPO               | 14.8    | 1.317 ± 0.102       | 0.497 ± 0.014     | 0.193 ± 0.009      | 3.96 ± 0.249       |
> | RLOO               | 15.56   | 1.280 ± 0.100       | 0.514 ± 0.014     | 0.192 ± 0.008      | 3.88 ± 0.243       |
> | Entropy            | 1.44    | 0.786 ± 0.073       | 0.267 ± 0.009     | **0.228 ± 0.009**  | 3.45 ± 0.193       |
> | Unlikely       | 10.04   | 1.381 ± 0.114       | 0.532 ± 0.015     | 0.191 ± 0.008      | 4.24 ± 0.239       |
> | BoN training    | **16.88** | 0.596 ± 0.055     | 0.541 ± 0.010     | 0.162 ± 0.008      | 2.29 ± 0.173       |
> | Ensembling          | -       | 1.143 ± 0.086       | 0.438 ± 0.014     | 0.211 ± 0.010      | 4.19 ± 0.269       |
> | MARA (rev)         | 15.42   | 1.451 ± 0.103       | 0.543 ± 0.014     | 0.186 ± 0.008      | 4.14 ± 0.233       |
> | MARA (fwd)         | 15.33   | **1.604 ± 0.113**   | **0.568 ± 0.012** | 0.193 ± 0.009      | **4.62 ± 0.258**   |
>
> ## Creative Q&A, ablation
>
> Ablation of threshold parameter $\tau$ for Section 6.2 (here is set as percentile of batch)
>
> | Model                   | Out Dist Reward (↑)   | EAD (↑)           | SemDiv (↑)      | Distinct (↑)    |
> |-------------------------|---------------------|--------------------|------------------|------------------|
> | MARA (rev, 0.9)         | 1.451 ± 0.103       | 0.543 ± 0.014      | 0.186 ± 0.008    | 4.14 ± 0.233     |
> | MARA (fwd, 0.9)         | 1.604 ± 0.113      | 0.568 ± 0.012  | 0.193 ± 0.009    |  4.62 ± 0.258    |
> | MARA (rev, 0.75)        | 1.498 ± 0.117       | 0.547 ± 0.013      | 0.183 ± 0.008    | 4.41 ± 0.262     |
> | MARA (fwd, 0.75)        | 1.325 ± 0.097       | 0.508 ± 0.014      | 0.196 ± 0.009    | 4.07 ± 0.243     |
>
>
> ## Drug discovery (Sec 6.3, Table 2)
>
> Ablation running MARA with $\tau = 0.85$. The Screening threshold is set a priori (in practice it is used as an initial filter to screen molecules for next step evaluation). Training MARA with higher $\tau $ results in further Yield improvement when the screening threshold is stricter.
>
>
> ### SYNTH task (Table 2a)
>
> | Screen Thresh | Algorithm             | Yield (↑)   | OB100 (↓)  | IntDiv1 (↑)    | Circles (↑) |
> |-------------|------------------------|-------------|------------|----------------|-------------|
> | 0.8         | REINVENT               | 6569±186    | 1042±66    | 0.766±0.011    | 67±3        |
> | 0.8         | MARA ($\tau = 0.80 $ )      | 6834±78     | 1015±55    | 0.761±0.009    | 59±8        |
> | 0.8         | MARA ($ \tau = 0.85 $ )      | 6584±231    | 1042±66    | 0.761±0.008    | 72±6        |
> ||||
> | 0.85        | REINVENT               | 1614±407    | 4114±109   | 0.701±0.018    | 7±1         |
> | 0.85        | MARA ($ \tau = 0.80 $ )      | 1796±210    | 3654±272   | 0.716±0.015    | 6±1         |
> | 0.85        | MARA ($ \tau = 0.85$ )      | 2196±394    | 4010±297   | 0.703±0.011    | 7±1         |
>
>
>
>
> ### ALL-AMIDE task (Table 2b)
>
> | Eval Thresh | Algorithm             | Yield (↑)   | OB100 (↓)  | IntDiv1 (↑)    | Circles (↑) |
> |-------------|------------------------|-------------|------------|----------------|-------------|
> | 0.8         | REINVENT               | 5433±184    | 1427±63    | 0.768±0.012    | 35±1        |
> | 0.8         | MARA ($ \tau = 0.80 $)      | 5635±249    | 1407±123   | 0.766±0.008    | 36±3        |
> | 0.8         | MARA ($ \tau = 0.85 $)      | 5502±309    | 1426±63    | 0.769±0.006    | 34±3        |
> ||||
> | 0.85        | REINVENT               | 1098±88     | 4360±257   | 0.721±0.016    | 8±1         |
> | 0.85        | MARA ($ \tau = 0.80 $)      | 1235±130    | 3943±303   | 0.733±0.009    | 8±1         |
> | 0.85        | MARA ($ \tau = 0.85 $)      | 1438±126    | 4230±401   | 0.725±0.008    | 8±1         |

---

> ### Author Response · Authors · 2025-11-26
> **Updated pdf with all mentioned changes**
>
> **We have updated the pdf** which addresses all changes as promised.
>
> To view the previous version (as the initial review and responses refer to line numbers in the first version), please refer to the initial submission on 25 Sept 2025, unde "_Revisions_".

---

### Author Response · Authors · 2025-12-03
**Summary of discussions**

We briefly summarize how the paper evolved during the review process and how we addressed the main concerns raised by the reviewers.

## Positive aspects highlighted in the initial reviews

- There is broad consensus the paper provides **timely understanding on a currently highly debated and important topic** regarding the behavior of KL-regularized policy gradient methods (W9ms, v1oM, K4Yn), and **provides insights that are often overlooked** in practice (mZ9a)
- The work identifies and analyzes an **important practical problem**: diversity collapse in KL-regularized RL as it is commonly applied to large generative models (v1oM).
- To clarify the behavior of the regularized RL objective, the paper makes **novel and non-trivial theoretical contributions** (mZ9a, K4Yn).
- Building on these insights, the paper proposes an **original, well-motivated algorithm** (W9ms, K4Yn) that modifies the reward to induce a more desirable target distribution, and evaluates it with **well-constructed experiments** (W9ms).
- The paper is **well written** (K4Yn), **enjoyable to read** (mZ9a), with **illustrations that effectively convey the key intuitions** (W9ms, v1oM).


## Main concerns in the initial round

- **Baselines for diversity**: lack of stronger diversity-promoting baselines for comparison.
- **Hyperparameter $\tau$**: questions about the need to set $\tau$, and the absence of an ablation to understand its effect.
- **Framing and clarity**: some confusion about the setting being RL for generative models (rather than standard MDP RL) and how this is framed in the paper.
- **Conceptual clarity around divergences and the target distribution**: (i) the relationship between the KL regularizer, the implicitly optimized divergence, and the shape of the resulting target distribution; (ii) wording in the abstract and introduction that appeared to conflate or blur these roles.
- **Terminology and definitions**: requests for more precise definitions (e.g., “modes”, “multimodal”) and a clearer discussion of related work.

## Revisions and new experiments

In our rebuttal and revision, we made the following changes:
- **Additional baselines for diversity**: we added stronger diversity-focused baselines in the Creative QA task (Section 6.2). MARA remains the best overall method relative to the added baselines.
- **Ablations and practical guidance on $\tau$**: added ablations on the effect of $\tau$ in both the Creative QA task (Section 6.2) and the drug discovery task (Section 6.3). These show that $\tau$ can be set in a straightforward way without harming performance, and the method is robust over a reasonable range of values.
- **Clarified problem setting and positioning**
  - We explicitly clarified early in the paper (L121) we focus on RL for generative models (contextual bandits), rather than generic MDP RL.
  - We also proposed a more precise title to better reflect this focus.
- **Improved conceptual and expository clarity**
  - We substantially rewrote the abstract and parts of the introduction to emphasize the paper’s goal as a didactic clarification of the role of regularization and the induced target distribution in KL-regularized RL.
  - We clearly separated the mode-seeking / mass-covering properties of minimizing reverse/forward KL, vs. the behaviour of reverse/forward KL _regularized_ RL, and avoided any phrasing that might overstate or oversimplify the “KL choice” as the sole cause of diversity collapse.
  - Specifically, we made explicit:
    - Current objectives construct target distributions that concentrate mass on a few high-reward regions, leading to mode collapse regardless of the specific algorithm used.
    - Changing the regularizer between reverse and forward KL (as commonly done in practice [1,2,3]) does not address this in a fundamental way.
    - All of this motivates our method, MARA, which uses reward augmentation to construct a target distribution which is designed to put mass uniformly over all high-rewarding regions, thus covering more modes in the reward, and therefore promotes diversity.
- **More precise definitions and terminology**: we refined the formal definition of “multimodal” in Definition 3.5 to mean “reward-multimodal”, referring to “all high-reward regions”. We also avoided using “mode” ambiguously in the abstract. Instead, we explicitly use “mass concentrating on a single high-reward region” as the operational definition for mode collapse.
- **Expanded related work discussion**: We added discussion of several related works in (i) diversity promoting methods, (ii) KL-regularized policy gradients from classical RL and max entropy RL.

---

> ### Author Response · Authors · 2025-12-03
>
> ## Additional discussion with reviewer K4Yn
>
> After the initial rebuttal, reviewer K4Yn engaged in a deeper discussion about the interpretation of our results, in particular: **Questioning whether the target distribution is the true culprit of diversity collapse**, asking for more evidence that the shape of the LLM base distribution contributes to target collapse in practice.
>
> In response to this discussion, we **ran additional diagnostic experiments to investigate the role of the pre-trained base policy** (added to Appendix C.4 of the updated manuscript)
>   - These experiments empirically show that the base policy assigns large $\log \pi_{\text{ref}}$ differences to samples with similar reward values, which induces a highly skewed (and effectively non-multimodal) target distribution.
>   - If we account for _both_ $\log \pi_{ref} $ and reward differences, a top response can be up to a factor of $10^{40}$ more likely than another response with only a slightly lower reward.
>   - Both results support the hypothesis that the induced target distribution for LLMs is **highly non reward-multimodal**
>
> We also further revised our abstract, with the major changes summarized in the section above.
>
> This exchange with reviewer K4Yn helped refine the paper’s messaging and reduce potential misinterpretations of our main claims.
>
> ## Overall
>
> All in all, we believe the final version of the paper addresses all concerns raised by the reviewers, strengthens the empirical and conceptual support for our claims, and improves the clarity of the exposition.
>
> We thank the reviewers for their thoughtful feedback. **We believe this work will be a meaningful contribution to the conference and provide valuable clarifying insights about regularized RL** (e.g. as commonly used for LLMs).
>
> -----
>
> [1] Wang, Chaoqi, Yibo Jiang, Chenghao Yang, Han Liu, and Yuxin Chen. 2024. BEYOND REVERSE KL: GENERALIZING DIRECT PREFERENCE OPTIMIZATION WITH DIVERSE DIVER- GENCE CONSTRAINTS.
>
> [2] Omura, Motoki, Yasuhiro Fujita, and Toshiki Kataoka. 2025. “Entropy Controllable Direct Preference Optimization.” arXiv:2411.07595. Version 2. Preprint, arXiv, June 13. https://doi.org/10.48550/arXiv.2411.07595.
>
> [3] Li, Long, Jiaran Hao, Jason Klein Liu, et al. 2025. “The Choice of Divergence: A Neglected Key to Mitigating Diversity Collapse in Reinforcement Learning with Verifiable Reward.” arXiv:2509.07430. Version 2. Preprint, arXiv, October 17. https://doi.org/10.48550/arXiv.2509.07430.

---

### Meta-Review · Area_Chair_nHeQ · 2026-01-06

**Summary:**

This paper analyzes KL-regularized reinforcement learning for generative models, arguing that diversity collapse is inherent to the objective's optimal solution rather than an optimization failure. The authors show that under common settings (low regularization strength, equal verifiable rewards), the target distribution concentrates mass on single high-reward regions regardless of whether forward or reverse KL regularization is used. They propose MARA (Mode Anchored Reward Augmentation), a simple reward modification that constructs target distributions with uniform probability across all high-reward regions.

**Reviewer Concerns:**

Addressed:
- Baselines and experiments
- Hyperparameter $\tau$ ablation
- Clarity on KL regularization vs. implicitly minimized divergence
- Evidence for unimodal targets in LLMs

**Reviewer Scores:**

The reviewers would have coincided onto a positive assessment, since the author rebuttal effectively address the raised concerns.

---

### Decision · Program_Chairs · 2026-01-26

Accept (Poster)